# Multi-level phenotypic models of cardiovascular disease and obstructive sleep apnea comorbidities: A longitudinal Wisconsin sleep cohort study

Duy Nguyen[1‡], Ca Hoang[2‡], Tien Truong[2], Dang Nguyen[3,4], Hillary Gia Lam[5], Abhay Sharma[6], Trung Quoc Le[2,4]*, Phat Kim Huynh[1,2]*

1 Department of Industrial and Systems Engineering, North Carolina A&T State University, Greensboro, North Carolina, United States of America, 2 Department of Industrial and Management Systems Engineering, University of South Florida, Tampa, Florida, United States of America, 3 Harvard T.H. Chan School of Public Health, Harvard University, Boston, Massachusetts, United States of America, 4 Department of Medical Engineering, University of South Florida, Tampa, Florida, United States of America, 5 College of Arts and Sciences, University of South Florida, Tampa, Florida, United States of America, 6 Department of Otolaryngology–Head and Neck Surgery, University of South Florida, Tampa, Florida, United States of America

‡ Co-first authors.
* pkhuynh@ncat.edu (PKH); tqle@usf.edu (TQL)

## Abstract

Cardiovascular diseases (CVDs) are prevalent among obstructive sleep apnea (OSA) patients, presenting significant challenges in predictive modeling due to the complex interplay of these comorbidities. Current methodologies predominantly lack the dynamic and longitudinal perspective necessary to accurately predict CVD progression in the presence of OSA. This study addresses these limitations by proposing a novel multi-level phenotypic model that analyzes the progression and interaction of these comorbidities over time. Our study utilizes a longitudinal cohort from the Wisconsin sleep cohort, consisting of 1,123 participants, tracked over several decades. The methodology consists of three advanced steps to capture the relationships between these comorbid conditions: (1) performing feature importance analysis using tree-based models to highlight the predominant role of variables in predicting CVD outcomes. (2) developing a logistic mixed-effects model (LGMM) to identify longitudinal transitions and their significant factors, enabling detailed tracking of individual trajectories; (3) and utilizing t-distributed stochastic neighbor embedding (t-SNE) combined with Gaussian mixture models (GMM) to classify patient data into distinct phenotypic clusters. In the analysis of feature importance, clinical indicators such as total cholesterol, low-density lipoprotein, and diabetes emerged as the top predictors, highlighting their significant roles in CVD onset and progression. The LGMM predictive models exhibited a high diagnostic accuracy with an aggregate accuracy of 0.9556. The phenotypic analysis yielded two distinct

**Data availability statement:** The data underlying the results presented in this study are sourced from the Wisconsin sleep cohort study (WSCS) on National Sleep Research Resource (NSRR) and available from https://sleepdata.org/datasets/wsc. WSCS data may be used for non-commercial use by those affiliated with an academic research institution only. Requests for data use will be approved exclusively for sleep related research. Additional variables and data may be requested and obtained directly from the WSCS (please contact Amanda Rasmuson: arasmuson@wisc.edu).

**Funding:** The author(s) received no specific funding for this work.

**Competing interests:** The authors have declared that no competing interests exist.

clusters, each corresponding to unique risk profiles and disease progression pathways. One cluster notably carried a higher risk for major adverse cardiovascular events (MACEs), attributed to key factors like nocturnal hypoxia and sympathetic activation. Analysis using t-SNE and GMM confirmed these phenotypes, which marked differences in progression rates between the clusters. In conclusion, our study provides a profound understanding of the dynamic OSA-CVD interactions, offering robust tools for predicting CVD onset and informing personalized treatment strategies.

## 1. Introduction

Obstructive sleep apnea (OSA) is a condition characterized by episodes of upper airway obstruction during sleep which is a prevalent yet commonly undiagnosed and undertreated comorbidity within the population of patients with cardiovascular diseases (CVDs) [1–4]. Alarmingly, 40% to 80% of OSA patients with CVDs, encompassing major adverse cardiovascular events (MACEs) categories [5]: (i) acute coronary syndrome/ischemic heart disease, (ii) chronic heart failure, (iii) cerebrovascular accidents, and arrhythmias, suffer from OSA. OSA-CVD comorbidity escalates overall morbidity and increases the risk of premature all-cause mortality [2,3,6–9]. Moreover, the coexistence of OSA and CVDs poses a considerable societal and economic burden, with healthcare-related expenses attributed to OSA reaching $150 billion and an extra $30 billion incurred when considering CVD-related comorbidities [10]. The interrelation between OSA and various cardiovascular diseases such as hypertension, coronary artery disease, and heart failure has been well-documented [11–17].

Numerous studies have rigorously explored the connections between OSA and CVDs. Particularly, OSA's hallmark feature of intermittent hypoxia has been linked to significant cardiovascular consequences, including systemic hypertension, atrial fibrillation, and heart failure, emphasizing the severity of its impact [2,6,12–14,18,19]. The mechanisms proposed to underlie this association include metabolic dysregulation [6,13,14], oxidative stress from intermittent hypoxia [2,18,19], and altered autonomic nervous system activity [2,14,19]. The intermittent hypoxia and reoxygenation in OSA activate systemic inflammatory pathways, leading to the release of proinflammatory cytokines which exacerbates endothelial dysfunction and vascular remodeling, creating a pro-thrombotic environment that heightens the risk of CVD [6,19]. Metabolic dysregulation in OSA patients often manifests as disrupted glucose and lipid profiles, particularly in individuals with coexisting obesity or diabetes [6]. Meanwhile, oxidative stress from intermittent hypoxia accelerates the progression of atherosclerosis by promoting lipid peroxidation, vascular smooth muscle proliferation, and plaque instability [20,21]. Sympathetic nervous system overactivity stimulated by the repetitive hypoxic episodes and arousals, coupled with reduced parasympathetic tone, also

contributes to the cardiovascular burden by sustaining elevated heart rate and blood pressure, thereby imposing greater cardiac workload and risk of myocardial injury [2,19,22,23]. However, the complexity of these interactions, compounded by a scarcity of longitudinal data, makes it challenging to untangle the temporal and causal relationships between these comorbid conditions [9,12,24]. This complexity is further magnified by the variability in individual patient responses and the multitude of confounding factors that obscure the direct impacts of these diseases on each other [11,25].

Despite extensive documentation of the OSA-CVD relationship, much of the existing literature relies on cross-sectional or observational studies, which can only infer associations rather than causal mechanisms. This has left significant gaps in our understanding of how these physiological disruptions progress over time to exacerbate or initiate CVD pathology in patients diagnosed with OSA. Longitudinal research probing the evolution and interaction of OSA and CVD remains scarce. Current literature predominantly employs generalized, cross-sectional models [26–30] that fail to capture endophenotype-specific dynamic physiological changes associated with OSA, such as intermittent hypoxia-induced oxidative stress and fluctuations in autonomic nervous system activity, which are key factors in the development and progression of CVD [2]. These gaps significantly hamper our capacity to predict critical disease transitions, thereby limiting timely interventions to prevent severe cardiovascular incidents. Furthermore, existing predictive longitudinal models for OSA-CVD comorbidity often oversimplify the complex OSA-CVD interactions, neglecting the multifactorial influences of genetic predispositions, environmental factors, and lifestyle choices [31,32]. As a result, such models do not adequately capture the heterogeneity of patient experiences and responses to treatment.

Moreover, the methodology commonly employed in studying these diseases seldom integrates advanced statistical techniques that could handle the dynamic complexity of multilevel biological and clinical data. The utilization of such methodologies is imperative not only for understanding how treatment regimens affect disease progression over time but also for identifying early indicators of disease exacerbation. For instance, while odds ratio or linear regression models effectively identify associations and predictive markers, they are inherently static and cannot explicitly handle the dynamic changes critical to understanding disease progression [26,28,32]. Similarly, more conservative approaches like survival analysis models capture certain longitudinal aspects of the OSA-CVD relationship but fail to account for repeated measurements or explore phenotypes in depth [33–36]. This leaves significant gaps in comprehensively modeling disease trajectories and the heterogeneity of patient experiences. There is a pressing need for endophenotype-specific predictive models that can incorporate time-varying covariates, account for inter-individual variability, and provide a robust model for simulating disease progression scenarios under various intervention strategies.

To overcome these limitations, our paper proposes a novel integration of logistic mixed-effects models (LGMMs) with advanced machine learning tools such as t-distributed stochastic neighbor embedding (t-SNE) and Gaussian mixture models (GMMs). This hybrid approach enables the dynamic analysis of CVD progression in OSA patients over time, bridging the gap left by static, cross-sectional methods. Our proposed LGMM can capture the temporal dynamics by modeling both fixed effects, which are consistent across all individuals, and random effects, which vary between individuals. This modeling structure depicts how CVD progression in OSA patients evolves from visit to visit, providing a dynamic overview rather than a static snapshot. By leveraging advanced feature engineering, our study also incorporates the multifactorial influences driving OSA-CVD interactions, ensuring a more comprehensive examination of drivers behind disease progression. The use of t-SNE for dimensionality reduction and GMMs for clustering allows the identification of distinct phenotypic clusters of patients. This methodology segments patients into groups with similar disease characteristics and progression patterns, effectively addressing endophenotypic variations within the population. Understanding these phenotypic patterns, shaped by genetic, environmental, and lifestyle factors, is critical for unraveling the diverse manifestations of CVD in OSA patients. In addition, our research methodology identifies and validates new predictive biomarkers and phenotypic patterns associated with CVDs in OSA patients.

 

## 2. Methodology

Our methodology for investigating the longitudinal associations between OSA and CVD within the Wisconsin sleep cohort study (WSCS) [37,38] comprises three steps, as illustrated in Fig 1. Initially, longitudinal data preprocessing and feature engineering are performed using the WSCS dataset. This stage involves categorical encoding, handling missing data, outlier detection, and synthetic minority over-sampling technique (SMOTE) for handling class imbalances by up-sampling the minority group by 100%. To address the missing data issue, we consider two approaches. Inverse probability weighting (IPW) adjusts observation weights based on missingness probabilities. However, its reliance on accurate probability estimation and the instability of weights, especially in longitudinal studies, limits its robustness [39–41]. In contrast, imputation techniques estimate missing values based on patterns in the data, offering computational efficiency and suitability for clinical datasets with random missingness. Given these factors, we adopt the k-nearest neighbors (KNN) imputation strategy. Initially, we exclude features with more than 30% missing values from the total dataset of 1080 rows (360 subjects × 3 visits) to ensure sufficient data for effective imputation. We then select the top 20 features using a tree-based feature ranking method. The second step employs LGMM to accommodate both fixed and random effects across patient timelines. Next,

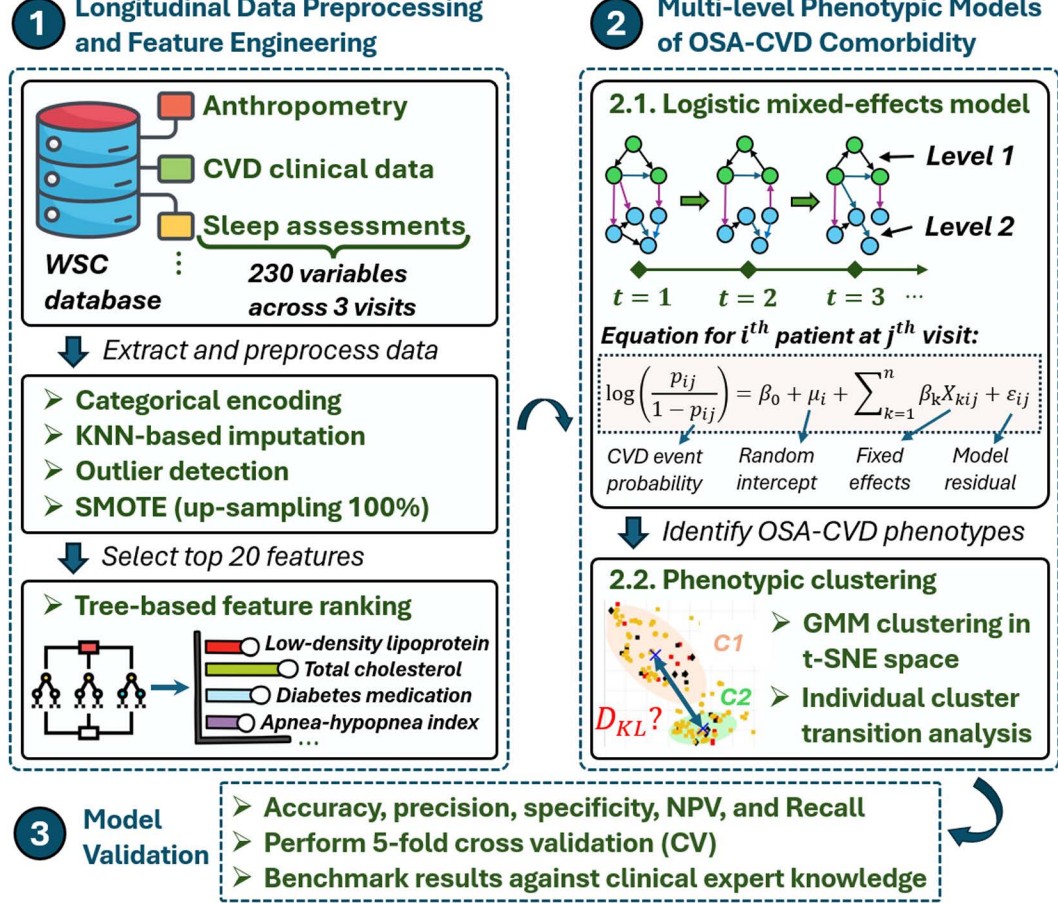

**Fig 1. Overview of the 3-step methodological framework for modeling OSA-CVD comorbidity utilizing WSC database.** Step 1 focuses on data preprocessing and feature engineering, step 2 details the LGMM and phenotypic GMM clustering, and step 3 outlines model validation approach.

we employ the t-SNE technique for dimensionality reduction and the GMM for OSA-CVD phenotypic clustering. Lastly, our model is validated using different evaluation metrics and expert clinical knowledge.

## 2.1. Wisconsin sleep cohort study description and our longitudinal study design

The study employs a longitudinal design utilizing data from the WSCS, which was initiated in 1988 to investigate the epidemiology and long-term health consequences of sleep disorders, with a particular focus on OSA. The WSCS includes comprehensive data collected from 1,500 initially recruited Wisconsin state employees. These employees were randomly selected to represent a wide demographic and health diversity, aged 30–85 at the time of recruitment. The WSCS encompasses 230 variables across multiple domains: anthropometry, clinical data, demographics, general health, lifestyle and behavioral health, sleep monitoring, sleep questionnaires, medical history, and sleep treatment. Each participant in the study was re-assessed at four-year intervals, with additional follow-ups to capture the progression and any new onset of health conditions.

The original WSSC design was planned for up to five follow-up visits per participant; however, the actual frequency and timing of these visits showed significant variability. Initial participation was robust with 1,123 participants at Visit 1, but subsequent visits saw a noticeable decline: 748 participants at Visit 2, 566 at Visit 3, 121 at Visit 4, and only 2 at Visit 5. This attrition and irregularity in follow-up compliance highlight the challenges in longitudinal health research. Our study strategically focuses on the first three visits, which offers more reliable data continuity. Furthermore, the intervals between visits varied widely, from 1 to 11 years. To address this, our analysis specifically targets 360 subjects who maintain regular follow-up intervals of 3, 4, or 5 years. Table 1 provides a structured overview of the sequence of CVD outcomes over three successive visits, based on the "any_cvd" variable, which indicates the presence of conditions such as heart attacks, congestive heart failure, and surgeries related to cardiovascular interventions.

A comprehensive comparative analysis of various health metrics categorized by CVD status across three successive visits within the WSCS is conducted. CVDs are classified into four MACEs [5] based on the self-reported presence of specific conditions and treatments: (1) MACE1 involves acute coronary syndrome/ischemic heart disease identified by factors such as heart attack and coronary artery conditions, (2) MACE2 pertains to chronic heart failure, (3) MACE3 includes arrhythmias with treatments like pacemakers, and (4) MACE4 represents nonfatal stroke.

## 2.2. Data preprocessing and feature engineering

Our data preprocessing involves four key steps: (1) categorical encoding, (2) KNN-based data imputation, (3) outlier detection, and (4) minority group over-sampling. First, the categorical variables are transformed into a numerical format using a label-encoding algorithm. To address missing data, the KNN-based imputation method is applied [42] to replace missing values by analyzing the similarities between data points. To identify outliers, a multivariate outlier detection

**Table 1. Longitudinal tracking of self-reported CVD outcomes in the WSCS database.**

| Group | Visit 1 (CVD Outcome) | Visit 2 (CVD Outcome) | Visit 3 (CVD Outcome) | Number of Patients |
|---|---|---|---|---|
| 0 | No CVD | No CVD | No CVD | 303 |
| 1 | No CVD | No CVD | CVD | 13 |
| 2 | No CVD | CVD | No CVD | 1 |
| 3 | No CVD | CVD | CVD | 9 |
| 4 | CVD | No CVD | No CVD | 4 |
| 5 | CVD | No CVD | CVD | 1 |
| 6 | CVD | CVD | No CVD | 0 |
| 7 | CVD | CVD | CVD | 29 |

approach is utilized using the Mahalanobis distance [43], which calculates the distance of a point from the dataset's mean, scaled by the covariance among variables.. Lastly, to enhance the LGMM's performance and address the class imbalance issue within our dataset, we utilized the SMOTE [44]. We opt for a 100% up-sampling of the *"CVD"* minority class, which ensures that less common but clinically significant patterns of CVDs are not overlooked, thereby improving the overall reliability and utility of our predictive LGMM.

In refining our feature engineering pipeline, we employ a tree-based method (including random forests or gradient boosting machines) that inherently evaluates the importance of each feature by measuring the reduction in impurity (typically Gini impurity or entropy) that each feature brings to the model [45]. Features that lead to significant splits in the tree are considered more important because they provide substantial information gain about the outcome variable. This process pinpoints which clinical, demographic, and physiological features are most crucial in predicting the progression of CVD in patients with OSA.

### 2.3. Multi-level phenotypic models of OSA-CVD comorbidity

To investigate the complex interplay between OSA and CVDs, the LGMM is developed to account for both fixed and random effects within longitudinal data. This model is specifically designed to handle variability both within and across individuals over time, enabling the study of complex interactions and transitions among patients with varying CVD profiles. To further refine our analysis and enhance the understanding of patient trajectories (i.e., sequences of data showing events and turning points of a disease condition), we employ t-SNE for dimensionality reduction, paired with GMM for clustering. This innovative approach allows us to visualize and categorize complex phenotypes within our data, grouping patients into distinct clusters that represent varying risks and progression patterns of CVD.

**2.3.1. Logistic mixed-effects model (LGMM).** The LGMM framework is designed to accommodate both fixed effects, which capture the general impact of observed variables across the entire population, and random effects, which allow for individual-specific variations. For instance, the fixed components of our model assess the average effects of known risk factors like age, body mass index (BMI), and smoking status on CVD incidence, whereas the random effects account for unobserved heterogeneity, potentially linked to genetic predispositions or lifestyle differences. Mathematically, the LGMM for our OSA-CVD analysis is structured as follows:

$$\log\left(\frac{p_{ij}}{1 - p_{ij}}\right) = \beta_0 + \varphi_i + \beta_1 X_{1ij} + \beta_2 X_{2ij} + \cdots + \beta_n X_{nij} + \varepsilon_{ij}$$

(1)

where:

- $p_{ij}$ is the probability of observing a cardiovascular event for the *i*-th subject at the *j*-th time point.

- $X_{kij}$ ($k = 1, \ldots, n$) is the *k*-th predictor for the *i*-th subject at the *j*-th time point. These predictors can represent various factors, including both OSA-related factors (*e.g.,* severity of nocturnal hypoxia and sleep fragmentation) and CVD-related factors (*e.g.,* cholesterol levels and blood pressure).

- $\beta_0$ represents the global intercept reflecting the baseline log odds of experiencing a CVD event.

- $\beta_k$ ($k = 1, \ldots, n$) are the coefficients for each predictor $X_{kij}$, indicating the impact of both OSA-related factors (*e.g.,* severity of nocturnal hypoxia, sleep fragmentation) and general health indicators (*e.g.,* cholesterol levels, blood pressure, and lifestyle factors).

- $\varphi_i$ denotes the random intercept for the *i*-th subject, capturing the unobserved heterogeneity due to individual susceptibility and resilience, which might influence the progression and outcomes of CVDs. These random effects are assumed to follow a normal distribution, $\varphi_i \sim N\left(0, \sigma_\varphi^2\right)$.

- $\varepsilon_{ij}$ is the error term, accounting for the residual variability not explained by the model.

The number of predictors $n$ is identified through a rigorous feature selection process. The specific value of $n$ and the corresponding predictors are chosen based on their statistical significance and contribution to the model's predictive performance, which are assessed through the tree-based feature importance analysis.

**2.3.2. High-dimensional patient trajectory tracking and phenotypic clustering with t-SNE and GMM.** Our study employs the t-SNE and GMM techniques to reduce the data dimensionality and to uncover latent structures and phenotypes that characterize the progression and variability of CVD among OSA patients. The t-SNE approach [46] provides a sophisticated means of visualizing the high-dimensional space of patient trajectories over multiple visits. Initially, t-SNE effectively reduces the dimensions of our dataset, which includes a broad range of clinical, biochemical, and demographic variables, into a two-dimensional space for visualization. Moreover, unlike other dimensionality reduction techniques, t-SNE preserves the local structure of the data, ensuring that similar data points in the high-dimensional space remain close in the reduced space.

After reducing dimensionality, we employ the GMM to identify distinct phenotypic clusters within the t-SNE transformed data. GMM provides a probabilistic means to model the data points as a mixture of multiple Gaussian distributions. Each component of the mixture represents a potential phenotype of OSA-CVD comorbidity, allowing us to classify patients into distinct groups based on their clinical profiles. GMM accommodates the heterogeneity within our patient data, modeling the distribution of each cluster with its own covariance structure. This flexibility allows for the effective handling of the diverse manifestations of CVD in OSA patients, ranging from mild to severe forms. We utilize the Expectation-Maximization (EM) algorithm to iteratively estimate the parameters of GMM, maximizing the likelihood function:

$$\theta = \arg\max_{\theta} \sum_{i=1}^{N} \log \left( \sum_{k=1}^{K} \pi_k \mathcal{N} \left( x_i | \mu_k, \Sigma_k \right) \right) \tag{2}$$

where $\theta$ represents the set of all parameters to be optimized in the GMM. These parameters include: (1) $\mu_k$: mean of the $k$-th Gaussian component, (2) $\Sigma_k$: covariance matrix of the $k$-th Gaussian component, and (3) $\pi_k$: mixing coefficient (or weight) of the $k$-th Gaussian component, representing the probability that a randomly selected datapoint belongs to this component. To further refine and validate our phenotypic clusters, we apply the Kullback-Leibler Divergence (KLD) [47], also known as relative entropy, to measure the distinctiveness of each phenotypic cluster's distribution compared to another one. This step ensures that the identified clusters are statistically significant and not artifacts of the dimensionality reduction process. The KLD between two continuous distributions $P$ and $Q$, with probabiltiy density functions $p(x)$ and $q(x)$, is defined as follows:

$$D_{KL} \left( P \| Q \right) = \int_{-\infty}^{\infty} p(x) \log \left( \frac{p(x)}{q(x)} \right) dx \tag{3}$$

### 2.4. Model validation

To rigorously assess the predictive performance of our models, we employ a comprehensive suite of statistical metrics, including Accuracy ($Acc$), Precision ($Pre$), Specificity ($Spec$), Negative Predictive Value ($NPV$), and Recall. Accuracy measures the overall effectiveness of the model in correctly predicting both CVD and non-CVD cases, providing a general assessment of model performance. It is calculated as the ratio of correctly predicted observations (true positives ($TP$) and true negatives ($TN$)) to the total number of cases, given by the formula: $Acc = (TP + TN)/(TP + TN + FP + FN)$. Here, $FP$ and $FN$ represent false positives and false negatives, respectively. Precision reflects the model's effectiveness in identifying only relevant instances among those retrieved. It is defined as the ratio of true positive predictions to all positive predictions: $Pre = TP/(TP + FP)$. Specificity ($Spec$), or the true negative rate, measures the proportion of actual negatives (non-CVD) correctly identified, an essential metric for confirming the absence of disease: $Spec = TN/(TN + FP)$. Negative

Predictive Value ($NPV$) indicates the likelihood that subjects not predicted to have CVD truly do not have the disease, enhancing confidence in model predictions of non-occurrence: $NPV = TN/(TN + FN)$. Recall, also known as sensitivity or the true positive rate ($TPR$), quantifies the model's ability to identify all relevant instances (actual CVD cases), crucial for ensuring that all potential CVD cases receive appropriate clinical attention: $TPR = TP/(TP + FN)$. The effectiveness of the models will be further evaluated through a five-fold cross-validation process, ensuring robustness and generalizability of the results across different subsets of data.

## 3. Results

This section presents the detailed outcomes of our analysis, systematically unfolding the complex interplay between OSA and CVDs within the WSCS. First, we evaluated the predictive power of various clinical indicators, using tree-based feature importance analysis to highlight key biomarkers that significantly influenced the forecasting of CVD progression. Subsequently, the LGMM performance was evaluated, demonstrating its capability to capture longitudinal transitions and provide robust statistical validation. Next, we explored phenotypic clustering using the t-SNE and GMM approaches, which revealed distinct CVD phenotypes through sophisticated dimensional reduction and clustering techniques. The validity of these phenotypic clusters was rigorously assessed using KLD, ensuring their distinctiveness and relevance in clinical contexts. Finally, comprehensive model validation metrics underscored the precision and reliability of our predictive LGMM.

### 3.1. Feature importance analysis

Prior to feature engineering, we conducted simulations to identify the optimal K for the KNN imputation method to handle missing data. Using a tree-based approach with five-fold cross-validation to calculate feature importance scores, we evaluated the performance of the KNN imputation method with K = 1, K = 3, and K = 5. The results indicate that for K = 3 and K = 5, the error bars of the importance scores vary significantly, indicating instability in the imputation. In contrast, K = 1 resulted in more stable and consistent results, suggesting that it is less prone to overfitting. Therefore, K = 1 was selected as the optimal parameter, as it provides reliable imputation while maintaining the integrity of the dataset. We also conducted a comprehensive comparison of the various health metrics arranged by CVD status. The results are documented in Table 2. Notably, significant differences are observed in metrics such as cholesterol levels, hypertension, and diabetes medication usage, suggesting their strong association with CVD progression in patients with sleep apnea. The detailed description of the variables in Table 2 is presented in S1 Table.

Our analysis utilized a tree-based method, specifically random forests [48], to rank the top 20 features across the WSCS dataset, integrating SMOTE and five-fold cross-validation to enhance the robustness and reliability of the results. Here, we employed SMOTE to address the significant class imbalance present in our dataset, where instances of non-CVD outcomes heavily outnumbered CVD outcomes. The SMOTE algorithm was configured to oversample the minority class (CVD cases) by 100%, effectively tripling the number of CVD cases in the training data. This oversampling was critical to provide a more balanced dataset, which helps in improving the classifier's ability to detect the minority class without overfitting. The Random Forest algorithm was utilized for feature ranking due to its efficacy in handling high-dimensional data and its robustness against overfitting. We configured the Random Forest with 100 trees. Each tree in the forest was allowed to grow to its maximum length without pruning, which permits the model to learn highly detailed patterns in the data. Feature importance was then derived from the average decrease in Gini impurity across all trees when splitting on a particular feature. The ranked features, as presented in Fig 2, revealed a pronounced emphasis on clinical data, with total cholesterol, LDL cholesterol, and diabetes medications scoring among top 20 of the most important features.

According to Fig 2, medical history variables, such as arthritis medication and health evaluations, showed significant relevance. Notably, general health measures, including serum creatinine levels, the demographic feature age, and anthropometric features such as waist circumference, were identified as key predictors. Furthermore, metrics from sleep

**Table 2. Comparative analysis of variables by CVD outcomes across three visits.**

| | Visit 1 | | Visit 2 | | Visit 3 | |
|---|---|---|---|---|---|---|
| | **No CVD (n = 326)** | **CVD (n = 34)** | **No CVD (n = 321)** | **CVD (n = 39)** | **No CVD (n = 308)** | **CVD (n = 52)** |
| **Anthropometry, mean(SD)** | | | | | | |
| bmi | 31.35(6.79) | 31.64(6.64) | 31.43(7.22) | 31.91(6.35) | 31.15(7.12) | 32.22(7.07) |
| waisthip | 0.9(0.09) | 0.97(0.1) | 0.91(0.1) | 0.97(0.08) | 0.93(0.09) | 0.98(0.1) |
| hipgirthm | 109.92 (14.13) | 106.48 (11.57) | 109.83 (14.77) | 107.51 (12.56) | 110.37 (14.66) | 107.85 (13.74) |
| neckgirthm | 38.75(4.18) | 40.75(4.47) | 38.36(4.09) | 41.07(4.08) | 38.15(3.97) | 40.25(4.52) |
| headcm | 56.73(2.41) | 57.46(2.18) | 56.62(2.08) | 57.94(1.55) | 56.58(2.02) | 57.63(2.02) |
| **Clinical Data, mean(SD)** | | | | | | |
| total_cholesterol | 203.16 (31.96) | 176.97 (32.35) | 197.97 (36.02) | 156.79 (22.09) | 187.25 (33.58) | 152.69 (35.7) |
| ldl | 121.64 (29.05) | 95.65 (31.73) | 114.96 (32.3) | 81.1 (22.83) | 107.49 (29.36) | 78.04 (26.29) |
| creatinine | 1 (0.18) | 1.03 (0.18) | 1 (0.2) | 1.06 (0.2) | 0.93 (0.24) | 1.05 (0.25) |
| triglycerides | 146.21 (80.75) | 160.24 (74.93) | 133.73 (78.59) | 127.15 (72.85) | 138.51 (74.91) | 149.56 (85.08) |
| uric_acid | 5.69 (1.35) | 6.05 (1.53) | 5.74(1.32) | 6.37(1.33) | 5.1(1.17) | 5.84(1.43) |
| sbp_mean | 125.47 (14.6) | 131.85 (12.34) | 126.34 (15.28) | 127.05 (13.85) | 127.07 (1.17) | 128.08 (1.43) |
| hdl | 52.88 (14.8) | 49.18 (15.00) | 57.05 (15.84) | 51.81 (14.58) | 52.83 (16.62) | 45.79 (13.58) |
| glucose | 103.31 (23.85) | 120.82 (49.33) | 106.17 (28.59) | 109.08 (20.03) | 101.26 (18.8) | 100.01 (19.73) |
| **Demographics** | | | | | | |
| sex, n(%) | | | | | | |
| *F* | 170(52.15) | 26(76.47) | 164(51.09) | 32(82.05) | 157(50.97) | 39(75) |
| *M* | 156(47.85) | 8(23.53) | 157(48.91) | 7(17.95) | 151(49.03) | 13(33.33) |
| age, mean (SD) | 55.97 (7.38) | 60.82 (6.91) | 59.94 (7.08) | 65.05 (7.23) | 64.36 (6.98) | 68 (7.75) |
| **General Health, n(%)** | | | | | | |
| eval_heath | | | | | | |
| *1: Excellent* | 52(15.95) | 5(14.71) | 51(15.89) | 3(7.69) | 40(12.99) | 6(11.54) |
| *2: Very good* | 163(50) | 10(29.41) | 157(48.91) | 11(28.21) | 14(48.38) | 10(19.23) |
| *3: Good* | 92(28.22) | 19(55.88) | 92(28.66) | 21(53.85) | 95(30.84) | 30(57.69) |
| *4: Fair* | 18(5.52) | N/A | 20(6.23) | 4(10.26) | 20(6.49) | 5(9.62) |
| *5: Poor* | 1(0.31) | N/A | 1(0.31) | N/A | 4(1.3) | 1(1.92) |
| **Lifestyle and Behavioral Health, n(%)** | | | | | | |
| caffeine | 2.74(2.23) | 3.06(2.77) | 2.57(2.16) | 2.67(2.17) | 2.56(2.05) | 3.1(2.45) |
| MACEs | | | | | | |
| *MACE1* | N/A | 29(85.29) | N/A | 34(87.18) | N/A | 45(86.54) |
| *treatMACE1* | N/A | 23(67.65) | N/A | 33(84.62) | N/A | 36(69.23) |
| *MACE3* | 41(12.58) | 14(41.18) | 37(11.53) | 17(43.59) | 39(12.66) | 17(32.69) |
| *treatMACE3* | N/A | 1(2.94) | N/A | 3(7.69) | N/A | 6(11.54) |
| *MACE2* | N/A | 1(2.94) | N/A | 4(10.26) | N/A | 3(5.77) |
| *MACE4* | 4(1.23) | 1(2.94) | 7(2.18) | 2(5.13) | 6(1.95%) | 3(5.77) |
| **Medical History, n(%)** | | | | | | |
| narcotics_med | 6(1.84) | 1(2.94) | 4(1.25) | 3(7.69) | 6(1.95) | 2(3.85) |

*(Continued)*

Table 2. (Continued)

| | Visit 1 | | Visit 2 | | Visit 3 | |
|---|---|---|---|---|---|---|
| | No CVD (n = 326) | CVD (n = 34) | No CVD (n = 321) | CVD (n = 39) | No CVD (n = 308) | CVD (n = 52) |
| anxiety_med | 26(7.98) | N/A | 22(6.85) | 1(2.56) | 23(7.47) | 4(7.69) |
| sedative_med | 25(7.67) | 2(5.88) | 29(9.03) | 4(10.26) | 39(12.66) | 11(21.15) |
| htn_med | 105(32.21) | 25(73.53) | 131(40.81) | 35(89.74) | 147(47.73) | 48(92.31) |
| arthritis_ynd | 95(29.14) | 13(38.24) | 113(35.2) | 16(41.03) | 129(41.88) | 27(51.92) |
| diabetes_med | 17(5.21) | 6(17.65) | 28(8.72) | 11(28.21) | 41(13.31) | 17(32.69) |
| **Sleep Monitoring, mean(SD)** | | | | | | |
| nremahi | 8.49 (14.04) | 13.16 (16.97) | 10.19 (13.69) | 14.39 (17.51) | 8.35 (12.48) | 14.51 (17.74) |
| ahi | 10.39 (14.56) | 15.21 (18.05) | 12.23 (14.29) | 16.61 (17.9) | 10.03 (13.1) | 16.4 (18.35) |
| avgo2sattst | 95.33(1.53) | 95.2(1.82) | 95.41(3.02) | 96.45(7.13) | 95.46(3.69) | 95.27(6.49) |
| pcttstrem | 17.11(5.94) | 16.66(5.71) | 16.47(5.88) | 15.85(6.65) | 15.15(5.96) | 14.3(6.72) |
| sleep_latency | 13.77 (17.32) | 9.68 (11.9) | 12.05 (13.08) | 10.95 (17.57) | 16.07 (18.12) | 12.51 (13.49) |
| **Sleep Questionnaires, n(%)** | | | | | | |
| apnea | 32(9.82) | 4(11.76) | 46(14.33) | 11(28.21) | 65(21.1) | 18(34.62) |
| **Sleep Treatment, n(%)** | | | | | | |
| apnea_treatment | 30(9.2) | 4(11.76) | 46(14.33) | 11(28.21) | 64(20.78) | 16(30.77) |

monitoring, specifically the average level of oxygen desaturation and mean oxygen saturation during total sleep, emphasized the interconnections between sleep quality and cardiovascular health.

## 3.2. Logistic mixed-effects model validation

Our evaluation of the LGMM extensively assessed its performance in predicting CVD outcomes (see Fig 3). This assessment utilized the top 20 features identified through the tree-based method, revealing significant model performance distinctions between LGMM and logistic regression (LR).

Here, an independent LR model was developed for each visit (Visit 1, Visit 2, Visit 3). Each model utilized the same top 20 features identified through the tree-based method that was employed for the LGMM. The LGMM and LR were trained on a dataset processed through five-fold cross-validation, a method chosen to mitigate overfitting and ensure the generalizability of the model predictions. Fig 3(a) details the aggregate performance metrics of both models across all data points pooled from the three visits. The LGMM demonstrated superior accuracy at 95.56%, which significantly surpassed the LR's 88.89%. In terms of precision, the LGMM achieved 80.21% compared to LR's significantly lower 24.75%. The recall metric further emphasizes the LGMM's robustness, with a score of 97.28%, indicating that it successfully identified nearly all actual CVD cases, whereas the LR managed only 56.39%. The precision disparity between the LGMM and LR stems largely from their structural differences and how they handle complex, longitudinal data. LGMM is specifically designed to include individual-specific random effects, which captures the unique trajectories of CVD progression among patients. This feature allows LGMM to more accurately identify true positive cases by accounting for both the fixed effects that are common across all subjects and the random effects that vary among individuals. In contrast, LR does not account for these intra-subject variations and repeated measures within the same subjects over time. This limitation can lead to a higher rate of false positives and false negatives, especially when the disease prevalence is low.

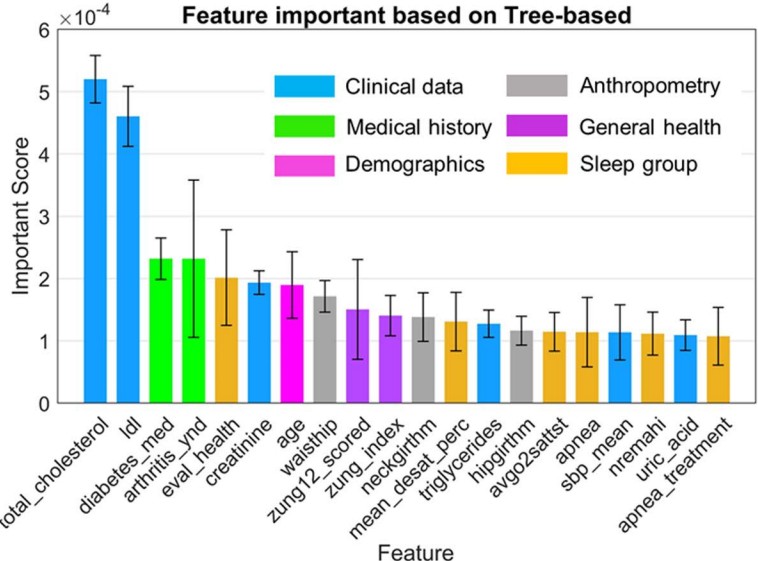

**Fig 2. Feature importance scores from tree-based model analysis.** This graph quantifies the predictive significance of each feature, with colors representing different data categories (*e.g.,* clinical data, medical history, demographics, and general health).

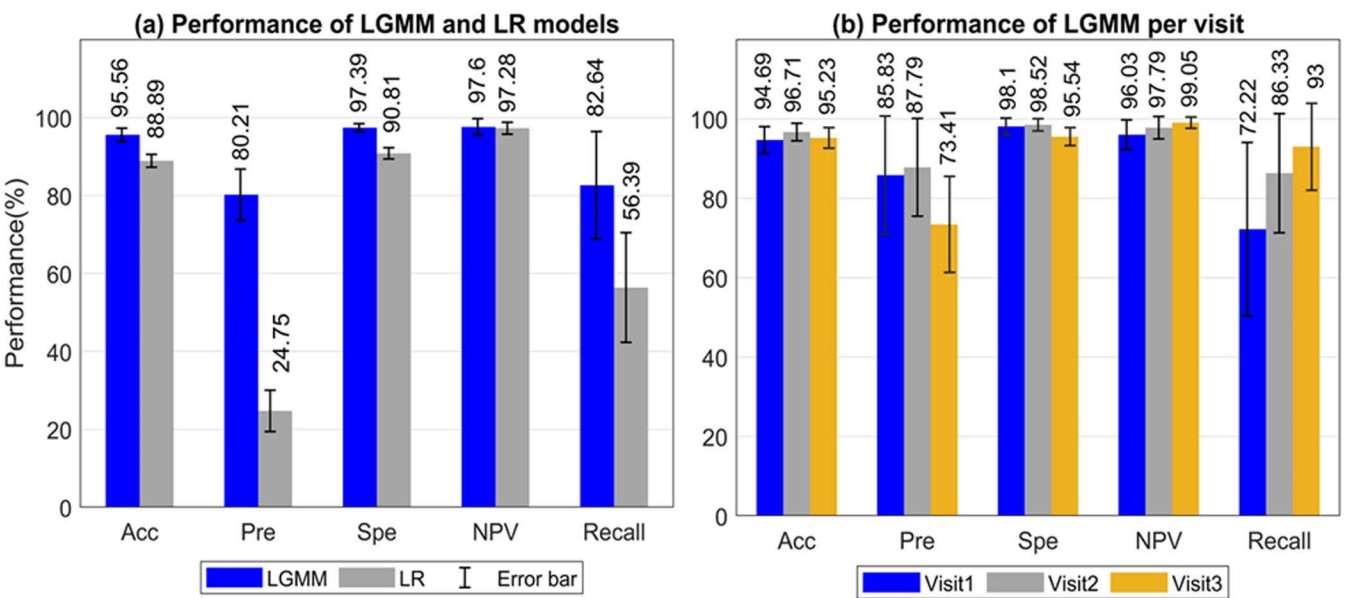

**Fig 3. A comprehensive evaluation of LGMM's performance across different scenarios.** Panel (a) shows the aggregate performance metrics of LGMM and LR models, highlighting the comparative performance based on the averaged values across three visits. Panel (b) details LGMM's performance per visit.

Fig 3(b) presents a detailed evaluation of the LGMM across three sequential visits within the study, revealing the model's sustained effectiveness in predicting CVD outcomes over time. The accuracy of the model consistently hovered around 95% for each visit. Specifically, the precision exhibited a modest decrease from 85.83% in the initial visit to

**Table 3. Estimated fixed effects of the representative variables from the LGMM analysis.**

| Variables | Estimate | SE | CI upper | CI lower | p-value | Sig code |
|---|---|---|---|---|---|---|
| wsc_vst | −0.0275 | 0.012 | 4.0797 | −1.9246 | 0.0224 | * |
| total_cholesterol | −0.0081 | 0.0027 | −0.0039 | −0.0511 | 0.0031 | ** |
| triglycerides | 0.0004 | 0.0004 | 0.0344 | −0.0184 | 0.245 | |
| sbp_mean | 0.0203 | 0.0062 | 0.3426 | −0.3433 | 0.001 | ** |
| hipgirthm | −0.013 | 0.0092 | 0.0037 | −0.0046 | 0.1583 | |
| zung_index | 0.0137 | 0.0077 | −0.0028 | −0.0135 | 0.0768 | + |
| diabetes_med | 0.0584 | 0.0345 | 0.0607 | −0.1032 | 0.0904 | + |
| nremahi | 0.0027 | 0.0016 | 0.0324 | 0.0082 | 0.0878 | + |
| avgo2sattst | −0.0246 | 0.0205 | 0.0051 | −0.0311 | 0.2306 | |
| mean_desat_perc | −0.0333 | 0.0218 | 1.5794 | −4.3209 | 0.1264 | |

Significance codes: p<0.001 '***', p<0.01 '**', p<0.05 '*', and p<0.1 '+'.

approximately 73.41% in Visit 3, potentially indicating slight variations in the model's positive predictive capacity as patient conditions evolve. The specificity and NPV values remained exceptionally high, both averaging close to 98%, which indicated the model's consistent ability to correctly identify non-CVD cases across different visits. Recall, or the model's sensitivity, rose gradually from around 72.22% during the first visit to 93% by the third visit. This incremental improvement suggests that the LGMM becomes increasingly effective at capturing true CVD cases.

In the detailed analysis of fixed effects within our LGMM, Table 3 presents the estimated fixed effects of key variables alongside their statistical significance.

As shown in Table 3, the variable *"wsc_vst"* (visit sequence) had an estimate of −0.0275 with a p-value of 0.0224, indicating a significant negative association with CVD outcomes, suggesting that later visits had a slightly lower risk of CVD, potentially reflecting some stabilization or effective management over time. Total cholesterol (*"total_cholesterol"*) showed a significant negative association with CVD outcomes (estimate: −0.0081, p-value: 0.0031). This counterintuitive finding likely reflects the use of cholesterol-lowering medications, such as statins, prescribed in response to high cholesterol levels. Statins reduce the risk of cardiovascular events and have been recommended as a primary prevention of CVD [49]. Systolic blood pressure (*"sbp_mean"*) showed a significant positive effect on CVD outcomes (estimate: 0.0203, p-value: 0.001), indicating that higher blood pressure was associated with increased CVD risk. Hip girth (*"hipgirthm"*) had an estimate of −0.013 with a p-value of 0.1583, indicating it was not statistically significant. The Zung self-rating depression scale index (*"zung_index"*) showed a borderline significant positive effect (estimate: 0.0137, p-value: 0.0768). Diabetes medication use (*"diabetes_med"*) was another important variable, showing a positive effect (estimate: 0.0584, p-value: 0.0904), indicating that individuals on diabetes medication had a higher risk of CVD, which aligned with the known comorbidity between diabetes and CVDs. For sleep-related variables, the average non-rapid eye movement apnea-hypopnea index (NREM-AHI, *"nremahi"*) showed a positive association with CVD risk (p-value: 0.0878), albeit marginally significant. However, other sleep metrics such as average oxygen saturation (*"avgo2sattst"*) and mean desaturation percentage (*"mean_desat_perc"*) did not show significant effects. In our detailed subgroup analyses, Table 4 presents the estimated fixed effects for representative variables within Group 1 and Group 3 defined in Table 1.

As observed in Table 4, Group 1 displayed significant findings in several variables. The variable *"wsc_vst"* had an estimate of 0.3678 with a p-value<0.001, indicating a strong positive association with CVD outcomes. This suggests that as the visits progressed, the likelihood of CVD diagnosis increased, reflecting the worsening health condition over time for these patients. Creatinine levels (*"creatinine"*) showed a significant positive association with CVD outcomes (estimate: 1.6643, p-value: 0.0088), indicating that higher creatinine levels, which often denote poorer kidney function, were linked

**Table 4. Estimated fixed effects of the representative variables for Group 1 and Group 3.**

| Variables | Estimate | SE | CI upper | CI lower | p-value | Sig code |
|---|---|---|---|---|---|---|
| **Group 1** | | | | | | |
| wsc_vst | 0.3678 | 0.0215 | 0.4102 | 0.3253 | < 0.0001 | *** |
| creatinine | 1.6643 | 0.6284 | 2.9046 | 0.424 | 0.0088 | ** |
| hdl | 0.0325 | 0.01 | 0.0522 | 0.0128 | 0.0013 | ** |
| ldl | −0.0113 | 0.0026 | −0.0062 | −0.0164 | < 0.0001 | *** |
| apnea_treatment | 0.06 | 0.0164 | 0.0923 | 0.0277 | 0.0003 | *** |
| anxiety_med | −0.2719 | 0.17 | 0.0636 | −0.6073 | 0.1115 | |
| sedative_med | 0.4033 | 0.1475 | 0.6944 | 0.1122 | 0.0069 | ** |
| sleep_latency | 0.0237 | 0.0065 | 0.0366 | 0.0108 | 0.0004 | *** |
| creatinine^2 | −0.9554 | 0.3134 | −0.3368 | −1.5741 | 0.0027 | ** |
| sleep_latency^2 | −0.0005 | 0.0002 | −0.0002 | −0.0009 | 0.005 | ** |
| **Group 3** | | | | | | |
| wsc_vst | 0.3758 | 0.0217 | 56.363 | −18.557 | < 0.0001 | *** |
| glucose | 0.0311 | 0.0063 | 0.4188 | 0.3327 | < 0.0001 | *** |
| ldl | −0.0092 | 0.0025 | 0.0435 | 0.0186 | 0.0004 | *** |
| headcm | −0.756 | 0.6527 | 0.0133 | −0.0202 | 0.2491 | |
| caffeine | 0.0693 | 0.0207 | −0.0042 | −0.0142 | 0.0011 | ** |
| pcttststage34 | 0.0841 | 0.013 | 0.5369 | −2.049 | < 0.0001 | *** |
| mean_desat_perc | 0.0544 | 0.0335 | 0.1103 | 0.0283 | 0.107 | |
| headcm^2 | 0.0069 | 0.0056 | −0.0001 | −0.0002 | 0.2261 | |
| caffeine^2 | −0.0111 | 0.0028 | 0.0002 | −0.0001 | 0.0001 | *** |
| pcttststage34^2 | −0.0043 | 0.001 | 0 | 0 | 0.0001 | *** |

Significance codes: p < 0.001 '***', p < 0.01 '**', p < 0.05 '*', and p < 0.1 '+'.

to increased CVD risk. HDL cholesterol showed a positive association with CVD outcomes (estimate: 0.0325, p-value: 0.0013). While HDL is traditionally considered cardioprotective due to its role in reverse cholesterol transport, recent studies suggest that dysfunctional HDL, characterized by impaired antioxidative and anti-inflammatory properties, may increase CVD risk in both healthy and clinical populations [50,51]. LDL cholesterol was negatively associated with CVD outcomes (estimate: −0.0113, p-value: < 0.001). While elevated cholesterol and LDL levels are widely recognized as risk factors for CVD, the reduced levels observed in certain clusters may reflect a more complex relationship between lipid profiles, disease progression, and treatment interventions. In patients with CVD, lower cholesterol levels might result from long-term statin therapy or other cholesterol-lowering treatments aimed at reducing cardiovascular risk [52]. The use of apnea treatment (*"apnea_treatment"*) was positively associated with CVD (estimate: 0.06, p-value < 0.001), indicating that those undergoing treatment for sleep apnea had a higher likelihood of CVD, perhaps due to the severity of their OSA condition necessitating treatment. Sedative medication (*"sedative_med"*) was positively associated with CVD (estimate: 0.4033, p-value: 0.0069), suggesting that sedative use might contribute to increased CVD risk, potentially due to its effects on sleep architecture. Sleep latency (*"sleep_latency"*) had a significant positive effect on CVD risk (estimate: 0.0237, p-value: 0.0004), indicating that longer time taken to fall asleep was associated with higher CVD risk. The quadratic term for creatinine (*"creatinine^2"*) showed a significant negative effect (estimate: −0.9554, p-value: 0.0027), highlighting a non-linear relationship where very high levels of creatinine might have different implications compared to moderate elevations. Similarly, the quadratic term for sleep latency (*"sleep_latency^2"*) also showed significance (estimate: −0.0005, p-value: 0.005), indicating a complex non-linear impact on CVD risk.

For Group 3, the *"wsc_vst"* variable had an estimate of 0.3758 with a p-value < 0.001. Glucose levels (*"glucose"*) were significantly positively associated with CVD outcomes (estimate: 0.0311, p-value: < 0.001), indicating that higher glucose levels were linked to increased CVD risk. LDL cholesterol showed a significant negative association (estimate: −0.0092, p-value: 0.0004), similar to Group 1. Caffeine intake (*"caffeine"*) was positively associated with CVD risk (estimate: 0.0693, p-value: 0.0011). The percentage of total sleep time in stage 3 and 4 sleep (*"pcttststage34"*) was significantly associated with CVD outcomes (estimate: 0.0841, p-value: < 0.001), suggesting that deeper stages of sleep had a notable impact on CVD risk. Mean desaturation percentage (*"mean_desat_perc"*) showed a positive but non-significant association (estimate: 0.0544, p-value: 0.107). Quadratic terms for head circumference (*"headcm^2"*), caffeine intake (*"caffeine^2"*), and sleep stages (*"pcttststage34^2"*) were included to capture non-linear effects.

The analysis of fixed effects between Group 1 and Group 3 highlighted nuanced differences in how clinical and lifestyle factors impact CVD outcomes in OSA patients. In Group 1, creatinine levels had a significant positive association with CVD outcomes. This group also exhibited a positive correlation between the use of apnea treatment and CVD risk, suggesting that more severe cases of OSA requiring intervention may inherently be at greater risk for cardiovascular complications. Intriguingly, the model for Group 1 showed that LDL cholesterol levels had a small negative association with CVD outcomes. This minimal effect size is likely attributable to confounding factors, such as the use of cholesterol-lowering therapies, rather than indicating a direct biological relationship. Additionally, the use of diabetes medications may confound these relationships. Such medications not only improve lipid profiles but also introduce anti-inflammatory and insulin-sensitizing effects, which could further complicate the interpretation of the observed associations between cholesterol levels and CVD outcomes [53].

Group 3 presented a different set of influential factors where glucose levels stood out. Additionally, the positive association of *"pcttststage34"*, representing deeper sleep stages, with CVD outcomes suggests that better quality sleep, involving more profound restorative stages, may offer protective benefits against CVD. The impact of lifestyle factors was distinctly noted in the relationship between caffeine consumption and CVD risk. The quadratic term for caffeine squared indicated an intricate relationship where moderate caffeine consumption could be benign or even beneficial, but excessive intake likely disrupts sleep architecture and thus increases CVD risk. Moreover, the analysis revealed that while certain variables like LDL cholesterol and caffeine had complex, nonlinear relationships with CVD outcomes, others such as creatinine and glucose levels showed more direct associations. This distinction suggests that metabolic and renal health directly influences cardiovascular risk, whereas the impact of lipid levels and lifestyle factors like caffeine consumption may depend significantly on their ranges and interaction with other risk factors.

### 3.3. Patient trajectory analysis of OSA-CVD comorbidity and phenotypic clustering

Utilizing the t-SNE and GMM approaches, we explored the patient transitions of these three groups across three clinical visits, as shown in **Fig 4**. This approach enabled us to track the progression and cluster formation based on phenotypic characteristics observed over time.

In investigating OSA-CVD comorbidity and phenotypic clustering, we concentrated on Groups 1, 3, and 7 from **Table 1** due to their distinct CVD development stages and their predominant sample sizes. We started by applying t-SNE to reduce the dimensionality of the dataset, which includes the top 20 features (illustrated in **Fig 2**) identified as the most predictive of CVD outcomes. The t-SNE algorithm was configured with a perplexity setting of 30, aiming to balance local and global aspects of the data, and an early exaggeration factor of 12 to ensure distinct clustering of the data points. We utilized the *Cosine* distance to measure the similarity between data points. Following the dimensionality reduction achieved with t-SNE, we employed GMM for clustering the reduced dataset. We determined the optimal number of clusters by evaluating the Bayesian Information Criterion (BIC) [54], ultimately choosing two clusters corresponding to the natural grouping suggested by t-SNE outputs. We set the "full" covariance type in GMM to allow each cluster its own general covariance matrix. The EM algorithm was used to estimate the model parameters. For accurate phenotypic clustering, we

identified and removed the outliers. This process ensured that our data set was homogenous, allowing for more precise and meaningful analysis. According to **Fig 4**, CVD patients were grouped into two distinct clusters based on a combination of clinical indicators and outcomes related to CVD and OSA. The detailed definitions of the two identified clusters are as follows:

- *Cardiovascular stability cluster* (Cluster 1): This cluster comprises patients who exhibit more favorable health outcomes. These individuals typically show lower average values of CVD risk biomarkers such as LDL cholesterol, fasting glucose, and fewer episodes of nocturnal hypoxia, suggesting a more controlled manifestation of both CVD and OSA. This cluster highlights a group with a slower progression of disease and more stable health profiles over time.

- *Cardiovascular risk cluster* (Cluster 2): Contrasting with Cluster 1, this group includes patients with poorer health outcomes, characterized by higher values of critical CVD risk biomarkers and more severe symptoms of sleep apnea. Patients in this cluster tend to demonstrate rapid progression of CVD and more severe manifestations of OSA, reflecting a higher overall health risk and a need for more aggressive treatment and management strategies.

The Davies-Bouldin Index and the average Silhouette Score were calculated to validate the derived clustering [55]. The Davies-Bouldin Index assesses cluster compactness and separation by comparing the similarity between the most similar

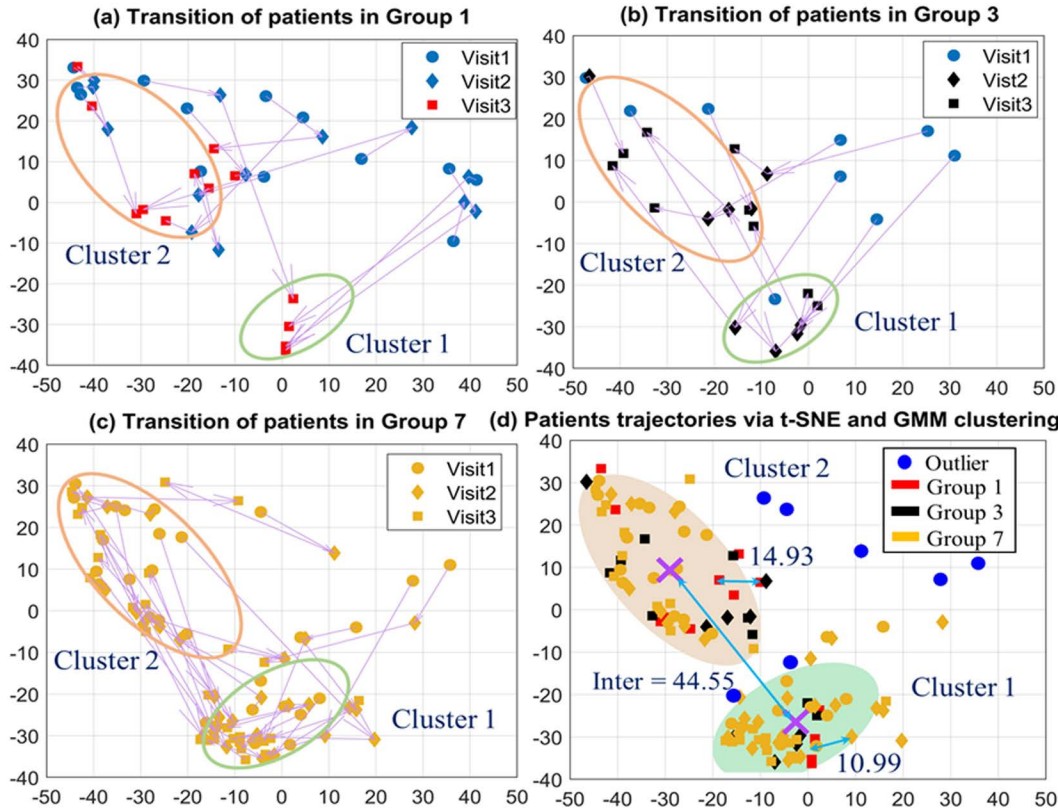

**Fig 4. Visualization of patient transitions and trajectories using t-SNE and GMM clustering, which illustrates the dynamic pathways of patients within Groups 1, 3, and 7 across three consecutive visits.** Panels (a)-(c) depict the trajectories of individual patients, mapped through visits 1 to 3, with transitions between two main clusters highlighted. The elliptical outlines in panels (a)-(c) represent Cluster 1 (green ellipse) and Cluster 2 (orange ellipse), indicating patient groupings based on their progression and health status. Panel (d) aggregates these transitions, showing the clustering of all three groups within the reduced dimensional space, and includes measures of intra- and inter-cluster Euclidean distances.

clusters, where a value close to 0 indicates well-separated and cohesive clusters [56]. Similarly, the Silhouette Score, ranging from −1–1, evaluates how well an object fits within its cluster compared to others. The average Silhouette Score of 0.867 and Davies-Bouldin Index of 0.567 of our clusters reflect strong cohesion within clusters and clear separation between them, confirming the effectiveness and reliability of the clustering approach. These results confirm that the identified clusters are both meaningful and reliable.

To further track the characteristics of patients with CVD in each group, we illustrated this in a tree diagram shown in **Fig 5**.

By examining the changes in characteristics of CVD patients across the three visits for Groups 1, 3, and 7, we can monitor whether a patient's condition falls into Cluster 1 or Cluster 2 and observe the specific attributes at each visit. For Group 7 observed in **Fig 4(c)**, it showed notable transitions both between and within clusters over the three visits. This group, characterized by long-term CVDs, exhibited significant within-cluster movement as well as shifts from one cluster to another. Notably, patients with CVD who were in Cluster 1 at Visit 1 consistently adhered to their treatment of diabetes through medication. These patients remained in Cluster 1 during the subsequent visits. Around 54% of patients in Cluster 2 transitioned to Cluster 1 by Visit 2, while 46% remained in Cluster 2. By Visit 3, the patients who had remained in Cluster 2 throughout the first two visits continued to stay in Cluster 2. Meanwhile, for patients who had been in Cluster 2 at Visit 1 and are now in Cluster 1 at Visit 2, 57% stayed in Cluster 1 at Visit 3, maintaining their health statuses, while 43% shifted back to Cluster 2. From Visit 1 to Visit 2, several patients in Cluster 1 move towards and even into Cluster 2, suggesting a change in their health status or a progression in disease severity. This movement continues into Visit 3, where additional transitions are observed not just into but also within Cluster 2, indicating ongoing changes in patient conditions.

The transition of patients via visits presented in **Fig 5(c)** illustrates the health trajectories of 24 patients in Group 7 across three visits. The Euclidean distances for transitions within clusters are relatively small, indicating stable health conditions. However, the transition distances between the two clusters across the visits, both from Cluster 1 to Cluster 2 and from Cluster 2 to Cluster 1, are significantly larger than the inter-cluster distances, highlighting substantial changes in health status. This indicates that patients moving between clusters experience more significant changes in their health than patients who remain in the same cluster. Furthermore, the presence of deceased patients in the second cluster underscores the severity of health deterioration compared to the first cluster. Patients in the second cluster generally exhibit more severe comorbidities and poorer health outcomes, as reflected by their higher values of C, T, A, and B.

Specifically, in Group 7, we divided the patients into four subgroups. Group 7.1 consists of 11 patients who maintained a stable health status throughout the visits. Group 7.1 starts with slightly elevated cholesterol (C = 158.73) and triglycerides (T = 99.91), but both metrics show a gradual decrease by Visit 3 (C = 134.22, T = 87.89). This indicates that these patients managed to maintain or slightly improve their health over time. The apnea-hypopnea index (AHI) and BMI values also remain stable, with A starting at 11.92 and slightly dropping to 9.67 by Visit 3, while B stays within the normal range, decreasing slightly from 31.27 to 28.42. Additionally, the number of patients on diabetes medication remains at D = 2 in both Visit 1 and Visit 2 but increases to 3 by Visit 3. Despite the overall stability in their health, the increase in diabetes medication usage suggests that some patients may have needed additional support in managing their diabetes as the study progressed. The number of patients with arthritis drops from Ar = 4–3, indicating a slight improvement in arthritis conditions. Group 7.2 consists of 4 patients, all women, who showed notable health improvement over time. At Visit 1, these patients were initially in Cluster 2, indicating poorer health conditions. However, by Visit 2, they transitioned to Cluster 1, signifying improved health, and they remained in this better health state through Visit 3. Specifically, at Visit 3, C = 123.50, T = 102.00, A = 29.43, B = 31.13, with 2 patients taking diabetes medication (D = 2) and 3 patients with arthritis (Ar = 3). This group highlights patients who responded well to treatment, likely improving their cholesterol, triglycerides, BMI, and AHI levels, while possibly managing comorbid conditions such as diabetes or arthritis. Notably, despite this positive shift, 2 patients in this group passed away. Group 7.3 comprises 3 patients, all women whose health status fluctuated over time. Initially, on Visit 1, they were in Cluster 2. On Visit 2, they experienced a temporary improvement,

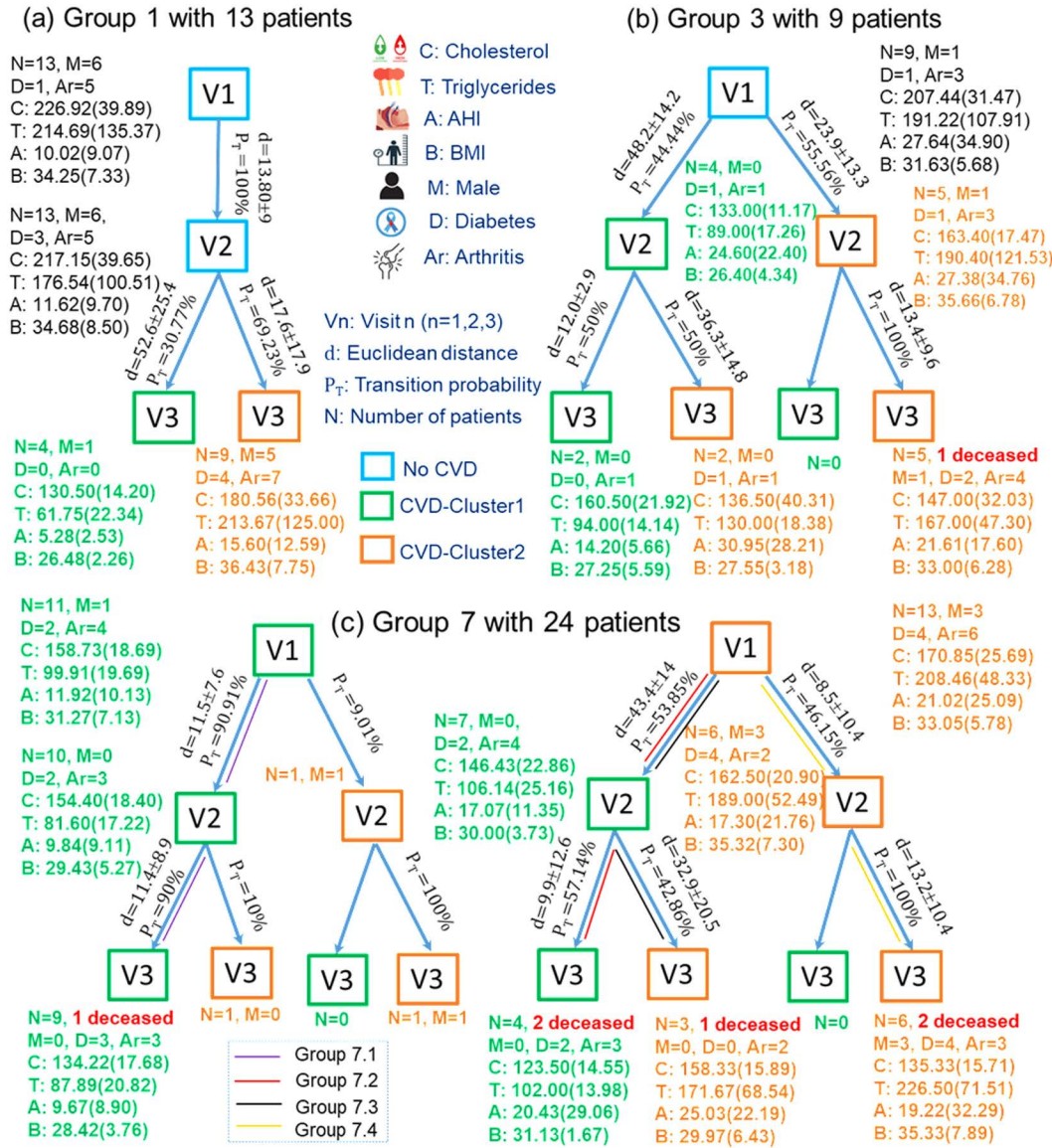

**Fig 5. Tree-based diagram of patient trajectories in CVD development.** The diagram tracks changes in critical health metrics—Total Cholesterol (C), Triglycerides (T), Apnea-Hypopnea Index (A), and Body Mass Index (B)—across three visits (V1, V2, V3) for Groups 1, 3, and 7. Each node within the trees represents a specific patient cohort per visit, detailing the total number of patients (N), the number of male patients (M), the number of patients with Diabetes medication (D), the number of patients with arthritis (Ar), mean values and standard deviations of the aforementioned metrics. The links between nodes depict Euclidean distances (d) indicating health metric variation over time and the estimated transition probabilities ($P_T$) that quantify the likelihood of progression between health states. Additionally, the nodes include the count of deceased patients within each group, reflecting mortality associated with CVD progression.

moving to Cluster 1. However, by Visit 3, their health deteriorated again, leading them back to Cluster 2. This group represents patients who struggled to maintain stable health status despite initial progress, indicating that they may have faced challenges in controlling critical health metrics such as cholesterol, triglycerides, and BMI. The fact that 1 patient in this group died highlights the difficulty of sustaining health improvements, especially for those with chronic conditions. At Visit 3, patients in Group 7.3 exhibited poorer health compared to those in Group 7.2, with higher values for cholesterol

(C = 158.33 vs. 123.50), triglycerides (T = 171.67 vs. 102.00), and slightly lower BMI (B = 29.97 vs. 31.13), while no patients required diabetes medication (D = 0) and 2 patients had arthritis (Ar = 2). Group 7.4 consists of 6 patients, 3 females who remained in Cluster 2 across all three visits, indicating consistently poor health. These patients did not show any improvement over time and likely had persistent difficulty managing key health indicators such as cholesterol, triglycerides, and BMI. Comorbidities such as diabetes medication and arthritis may have further contributed to their inability to transition to a better state of health. At Visit 2, compared to patients in Cluster 1 (including Groups 7.2 and 7.3), Group 7.4 showed poorer health, with D = 4, Ar = 2, C = 162.50, T = 189.00, A = 17.30, and B = 35.32. In contrast, Group 7.2 had D = 2, Ar = 4, C = 146.43, T = 106.14, A = 17.07, and B = 30.00. A similar trend was observed at Visit 3, where patients in Group 7.4 continued to exhibit poorer health outcomes than those in Groups 7.1, 7.2, and 7.3, as shown in **Fig 5(c)**. Therefore, patients in Group 7.4 experienced severe and persistent health issues throughout the study. Their inability to improve key health metrics, coupled with the presence of comorbidities, placed them at a significantly higher risk of negative outcomes. This is evidenced by the fact that 2 out of the 6 patients passed away, highlighting the grave health challenges and the critical need for more intensive interventions. Given the limited sample size, these findings should be considered exploratory and indicative of potential areas for further research rather than definitive conclusions.

In Fig 4(a) for Group 1, across three visits spanning 6–10 years, the patients transition from initial dispersion to progressive convergence into two distinct clusters. By the third visit, these clusters were well-defined, with patients in Cluster 1 displaying a consistent downward transition towards CVD manifestation compared to more complex trajectories in Cluster 2. This consistent transition in Cluster 1 might suggest either a more effective management of risk factors or a later onset of critical CVD symptoms. Patients in Cluster 1 show significant improvements in health metrics (C = 130.50, T = 61.75, A = 5.28, B = 26.48) compared to the previous visit (C = 217.15, T = 176.54, A = 11.62, B = 34.68), indicating a healthier profile with normal cholesterol and triglycerides levels, mild AHI, and overweight BMI (as shown in S2 Table). Moreover, none of the patients received treatment for diabetes nor were they diagnosed with arthritis (D = 0, Ar = 0), suggesting that all patients who were previously treated with diabetes and diagnosed with arthritis (D = 3, Ar = 5) have entered Cluster 2. On the other hand, Cluster 2 patients exhibit increased health metric values (T = 213.87, A = 15.60, B = 36.43) compared to the previous visit, except for total cholesterol which is markedly higher than in Cluster 1 (C = 180.56 vs. 130.50). More patients in Cluster 2 require diabetic treatment and receive arthritis diagnosis compared to the previous visit (D = 4, Ar = 7), reflecting more severe comorbidities and poorer health outcomes. Despite cholesterol levels remaining in the desirable range, patients in Cluster 2 present with high triglyceride level, moderate AHI, and obese-level BMI (S2 Table). Additionally, the proportion of male patients in Cluster 1 was significantly lower than that in Cluster 2 (0.25 and 0.56, respectively).

For Group 3 (Fig 5(b)), the trajectory analysis from Visit 1 to Visit 3 shows a rapid convergence into two clusters by the second visit, reflecting a faster progression of CVD. This group's patients move from a pre-CVD state at Visit 1 to a definite CVD state by Visit 2. Those transitioning to Cluster 2 by Visit 2 likely exhibit more severe comorbidities or a higher accumulation of risk factors, leading to deteriorated health outcomes by Visit 3. The trajectory data might suggest that patients in Cluster 1 of this group maintain a relatively healthier profile, while those in Cluster 2 develop more severe comorbidities, as shown by shorter distances indicating a quicker convergence to adverse health outcomes, shown in Fig 5(b). The characteristic data (C = 133.00, T = 89.00, A = 24.60, B = 26.40) imply that patients in Cluster 1 of this group maintain a relatively healthier profile at Visit 2, with the reported metric values corresponding to desirable cholesterol and triglycerides levels, but moderate AHI and overweight BMI (S2 Table). In contrast, those in Cluster 2 develop more severe comorbidities, evidenced by higher clinical metric values at Visit 2 (C = 163.40, T = 190.40, A = 27.38, and B = 35.66) and a quicker convergence to adverse health outcomes by Visit 3, including the presence of one deceased patient. The categories that the C and A metrics of Cluster 2 fall into are similar to those in Cluster 1 despite higher values. However, the average triglyceride level in Cluster 2 (T = 190.40) falls into the borderline high category, whereas the BMI (B = 35.66) indicates obesity (S2 Table). Across both clusters, there was only one male patient who remained in Cluster 2 throughout the visits, which may suggest that male patients are predisposed to greater cardiovascular risks than female patients.

In panel (d) of **Fig 4**, the patients' trajectories across Groups 1, 3, and 7 with CVD are visualized using t-SNE combined with GMM clustering. Specifically, Group 1 includes Visit 3, Group 3 includes Visits 2 and 3, and Group 7 includes all 3 visits. This method distinctly categorized these trajectories into two primary clusters. This intra- and inter-cluster mobility suggests fluctuations in individual patient health parameters, potentially driven by variations in treatment efficacy, changes in lifestyle, or the natural progression of their disease. The distance metrics between visits within clusters further suggest variability in how CVD impacts these patients over time. The two clusters are analyzed in detail in **Table 5**, where we evaluate the differences in clinical, demographic, and sleep-related variables between the two clusters, utilizing hypothesis testing to establish the statistical significance of the observed differences.

In **Table 5**, hypothesis testing was performed tailored to the data type of each variable. The differences in categorical variables were analyzed using Chi-square tests or Fisher's exact tests, and the continuous variables underwent two-sample t-tests to compare means between the clusters. Our analysis revealed that Cluster 1 exhibited significantly higher rates of treatment for MACE1, with 90.91% of patients receiving treatment compared to only 66.07% in Cluster 2. This difference, approaching statistical significance with p-value = 0.0836, suggests that Cluster 1 may benefit from more aggressive or effective CVD management strategies. Anthropometric measures demonstrated clear distinctions between the clusters. Hip-girth measurements were significantly larger in Cluster 2 with a p-value < 0.0001, suggesting a potential correlation with increased CVD risk often associated with higher body mass. Additionally, a BMI p-value < 0.0001 highlights the influence of BMI variations on the cardiovascular profiles of patients in these clusters. The clinical data highlighted substantial differences between the clusters, particularly in markers indicative of metabolic health. Creatinine and LDL were both significant with p-values of 0.0001 and 0.0281, respectively. Furthermore, total cholesterol and triglycerides were markedly higher in Cluster 2 (p-values < 0.0001 for both), pointing to a disturbed metabolic profile which is closely linked to increased CVD risk. These findings suggest that individuals in Cluster 2 might be at a higher metabolic risk, which could predispose them to more severe cardiovascular conditions.

The analysis of demographic factors revealed a significant divergence in sex distribution and age between the clusters. In Cluster 1, only 3.64% of individuals were male, whereas in Cluster 2, this proportion was higher at 35.71%, with a highly significant p-value < 0.0001. Age also differed significantly, with Cluster 1 being older on average than Cluster 2 (p < 0.0891). Significant disparities were also observed in medical history, particularly in the usage of hypertension medication, diabetes medication and the prevalence of arthritis. The use of hypertension medication and the prevalence of arthritis were notably higher in Cluster 2 (p < 0.0149 and p < 0.012, respectively). This indicates that hypertension may be more prevalent or more aggressively treated in this group. Additionally, the higher prevalence of arthritis in Cluster 2 suggests that individuals in this cluster may experience more musculoskeletal issues, leading to a greater burden of joint pain and mobility limitations. Differences in OSA metrics were statistically significant (p = 0.0463). This highlights the critical need for effective management of sleep disorders as part of comprehensive CVD risk reduction strategies.

**Fig 6** presents a comparative analysis of the distribution distances between Cluster 1 and Cluster 2, utilizing KL divergence $D_{KL}$ to quantify the disparities in feature distributions across these clusters.

Clinical data such as *"triglycerides"* showed the most pronounced divergences, with $D_{KL} \approx 3$, suggesting significant disparities in the metabolic profiles of patients in the two clusters, potentially reflecting different stages or intensities of metabolic complications related to cardiovascular health. *"total_cholesterol"* and *"ldl"* also showed notable divergences with $D_{KL}$ values around 0.75, indicating substantial differences in lipid profiles between the clusters. Sex within the demographics category showed moderate divergences with $D_{KL}$ values around 0.6, suggesting varying distributions that could influence the clusters' risk profiles and treatment responses. General health scores, particularly *"zung12_scored"* and *"zung_index"*, showed lower but noticeable $D_{KL}$ values, indicating differing levels of depressive symptoms between the clusters. Anthropometric features like *"bmi"* showed a highly significant $D_{KL}$ value of over 3, reflecting a pronounced difference in body composition between the clusters. Other anthropometric features such as *"hipgirthm"* exhibited $D_{KL}$ around 0.6–0.7, also indicating significant differences. Sleep-related variables such as *"nremahi"*, *"ahi"* and *"avgo2sattst"* showed lower $D_{KL}$

**Table 5. Statistical significance of the differences in MACE outcomes, anthropometric measures, clinical data, demographic factors, medical history, and sleep monitoring metrics between two clusters.**

| Categories | Variables | Cluster 1 (n = 55) | Cluster 2 (n = 56) | p-value | Sig code |
|---|---|---|---|---|---|
| **MACEs, n(%)** | MACE1 | 53(96.36%) | 47(83.93%) | 0.0526 | + |
| | MACE1 treatment | 50(90.91%) | 37(66.07%) | 0.0836 | + |
| | MACE3 | 21(38.18%) | 21(37.5%) | 0.0122 | * |
| | MACE3 treatment | 3(5.45%) | 4(7.14%) | 1 | |
| **Anthropometry, mean(SD)** | hipgirthm | 102.63(8.97) | 113.28(13.78) | < 0.0001 | *** |
| | BMI | 29.47(5.1) | 34.9(6.8) | < 0.0001 | *** |
| | waisthip | 0.98(0.08) | 0.98(0.1) | 0.7241 | |
| | neckgirthm | 40.45(3.51) | 41.38(4.52) | 0.2333 | |
| **Clinical Data, mean(SD)** | total_cholesterol | 145.6(21.82) | 165.39(29.7) | 0.0001 | ** |
| | ldl | 74.91(19.71) | 84.06(23.43) | 0.0281 | * |
| | creatinine | 1.11(0.21) | 1.01(0.22) | 0.0246 | * |
| | triglycerides | 90.35(22.26) | 205.38(79.83) | < 0.0001 | *** |
| | sbp_mean | 130.61(13.6) | 128.88(12.02) | 0.4802 | |
| **Demographics** | sex: M, n(%) | 2(3.64%) | 20(35.71%) | < 0.0001 | *** |
| | age, mean(SD) | 67.15(6.83) | 64.7(8.14) | 0.0891 | + |
| **Medical History, n(%)** | narcotics_med | 3(5.45%) | 3(5.36%) | 1 | |
| | htn_diuretic_med | 15(27.27%) | 28(50%) | 0.0149 | * |
| | arthritis_ynd | 19(34.55%) | 32(57.14%) | 0.0120 | * |
| | diabetes_med | 13(23.64%) | 19(33.93%) | 0.0302 | * |
| **Sleep Monitoring, mean(SD)** | apnea | 12(21.82%) | 16(28.57%) | 0.0463 | * |
| | nremahi | 11.7(12.31) | 16.59(20.61) | 0.1329 | |
| | ahi | 13.39(12.69) | 19.31(21.63) | 0.0818 | + |
| | avgo2sattst | 95.01(3.42) | 95.73(7.18) | 0.5041 | |
| | pcttstrem | 15.11(6.52) | 15.26(6.71) | 0.9075 | |

Significance codes: $p < 0.001$ '***', $p < 0.01$ '**', $p < 0.05$ '*', and $p < 0.1$ '+'.

values around 0.6–1. However, these differences were still critical in the OSA-CVD comorbidity context, suggesting that while there were differences in OSA severity, they were less pronounced than other features.

We have also conducted a detailed analysis of subjects' transition based on Fig 5, focusing on cholesterol medication monitoring in the groups developing CVD and with long-term CVD. S3 Table shows the analysis of patients in the pre-CVD stage at Visit 2. At this stage, patients tend to progress into one of the two identified CVD states: Cluster 1 (cardiovascular stability cluster) or Cluster 2 (cardiovascular risk cluster). We defined G1V2HV3C1 as patients at Visit 2 likely to progress towards Cluster 1 at Visit 3, and G1V2HV3C2 as patients at Visit 2 who tend to move towards Cluster 2 at Visit 3. Overall, patients in G1V2HV3C1 showed a better response to cholesterol treatment, maintaining a healthier profile with fewer comorbidities and lower AHI levels, supporting their classification as CVD-healthy in Cluster 1. In contrast, G1V2HV3C2 patients experienced more health complications, with higher comorbidity rates, limited improvement in lipid profiles, and worsening triglyceride levels, indicating a decline in cardiovascular health outcomes in Cluster 2.

In group **G1V2HV3C1**, all patients were in a healthy status at Visit 2 and received no cholesterol treatment despite elevated levels of cholesterol (236.75 mg/dl), LDL (160.25 mg/dl), and triglycerides (111.75 mg/dl). By Visit 3, all patients have started cholesterol medication, resulting in significant reductions in lipid levels to 130.50 mg/dl, 71.75 mg/dl, and 61.75 mg/dl, respectively. AHI levels remained low across the pre-CVD stage and the CVD stage in Cluster 1. Between Visit 2 and

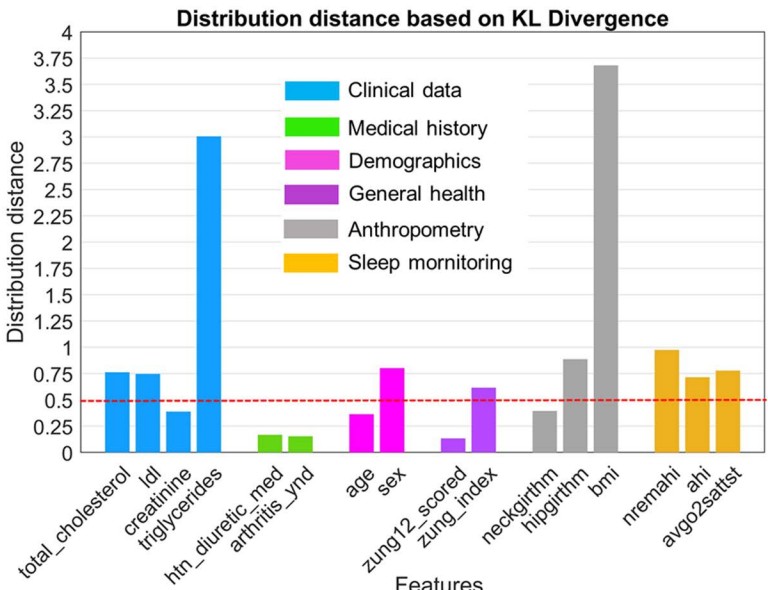

**Fig 6. Distribution distances between Cluster 1 and Cluster 2 based on Kullback-Leibler divergence.** The plot quantifies the disparity in the distribution of key features categorized by clinical data, medical history, demographics, general health, anthropometry, and sleep monitoring. The red dashed line in the figure represents a threshold $D_{KL}$ value of 0.5, highlighting features with substantial distribution differences.

Visit 3, the mean AHI increased from 4.05 to 5.28, while the NREM-AHI rose from 3.95 to 4.83. At Visit 3, the patients transitioned to the CVD stage in Cluster 1, with 75% receiving CVD treatment.

In group G1V2HV3C2, at Visit 2, 55.56% of patients without cholesterol treatment exhibited higher levels of cholesterol (224.50 ± 53.16 mg/dl), LDL (130.20 ± 58.83 mg/dl), and triglycerides (258.60 ± 115.33 mg/dl) compared to the those receiving treatment (cholesterol: 189.00 ± 21.37 mg/dl, LDL: 140.50 ± 16.74 mg/dl, triglycerides: 138.75 ± 4.50 mg/dl). Both groups shared similar constitutions, including comorbidities such as diabetes, arthritis, and OSA. They also displayed mid-obstructive to moderate AHI levels and elevated BMI levels, classifying many individuals as obese. These factors suggest that both groups face an increased risk of developing CVD over time, with a tendency towards poorer health outcomes. By Visit 3, all patients transitioned to Cluster 2, with 77.78% receiving cholesterol treatment. Among the treatment group, although there was a slight reduction in cholesterol levels (189.00 to 169.00 mg/dl) and LDL levels (104.05 to 89.29 mg/dl), triglyceride levels significantly increased from 138.75 to 172.43 mg/dl. For the 22.22% not receiving cholesterol treatment, cholesterol and LDL levels showed little change, but triglycerides rose significantly from 258.60 to 358.00 mg/dl, underscoring treatment's impact on lipid levels. These trends, also observed in S4 Table for the two sub-groups: G3V1HV2C1 (patients with no CVD at Visit 1 but had CVD in Cluster 1 at Visit 2) and G3V1HV2C2 (patients with no CVD at Visit 1 but had CVD in Cluster 2 at Visit 2), suggest that treatment improves lipid metabolism, potentially benefiting cardiovascular health. Therefore, the reduced cholesterol and LDL levels can be considered confounding factors.

S5 and S6 Tables present the analysis of patients in the long-term CVD group (Group 7), comparing the following categories: patients who transitioned from Cluster 2 to Cluster 1 (G7V1C2V2C1), patients who remained in Cluster 2 across both visits (G7V1C2V2C2), patients who transitioned from Cluster 1 to Cluster 2 (G7V2C1V3C2), and patients who remained in Cluster 1 across both visits (G7V2C1V3C1). S5 Table focuses on the sub-groups G7V1C2V2C1 and G7V1C2V2C2, while S6 Table focuses on sub-groups as G7V2C1V3C2 and G7V2C1V3C1. In S5 Table, both sub-groups received cholesterol treatment, resulting in similar lipid levels except for triglycerides, which showed a significant difference. For individuals transitioning from Cluster 2 to Cluster 1, the triglyceride level was 179.67 md/dl, while for those

remaining in Cluster 2, it was significantly higher at 232.17 mg/dl. A similar pattern was observed for NREM-AHI and AHI levels: transitioning patients had lower NREM-AHI (14.80) and AHI (18.48) compared to those remaining in Cluster 2 (NREM-AHI: 21.18, AHI: 23.18). This suggests that transitioning from Cluster 2 to Cluster 1 is associated with improvements in lipid and AHI, reflecting a potential shift towards better overall health. Conversely, individuals remaining in Cluster 2 exhibit persistently higher triglycerides and worse respiratory parameters, highlighting continued health risks.

In S6 Table, all patients received cholesterol treatment. Patients who remained in Cluster 1 had lower AHI, NREM-AHI, and BMI values at Visit 2 compared to Visit 3. This indicates that failure to maintain these metrics within a healthy range increases the risk of transitioning to Cluster 2, a poorer health state. Individuals transitioning to Cluster 2 exhibited higher lipid and AHI levels compared to those remaining in Cluster 1. This suggests that maintaining these indicators at lower levels, as observed in patients transitioning from Cluster 2 to Cluster 1 in S5 Table, is associated with better overall health. Conversely, an increase in these indicators among individuals remaining in Cluster 2 or moving towards it could be a sign of deteriorating health risk, particularly related to respiratory disorders and lipid metabolism. This highlights the crucial role of medical interventions in controlling these indicators to prevent health decline.

## 4. Discussion

Our paper presents a comprehensive exploration of OSA-CVD phenotype-based predictive modeling over a longitudinal timeline. The extensive data collected from the WSCS provides a robust foundation for our analysis, allowing for an intricate dissection of the progressive nature of these comorbidities through innovative statistical and machine learning techniques. Feature engineering, particularly through advanced tree-based methods, allowed us to distill the most relevant predictors from complex and high-dimensional data. This not only improved model accuracy but also enhanced our understanding of the key factors contributing to CVD risks in OSA patients. By identifying top features such as AHI, levels of oxygen desaturation during sleep, and various lipid profiles, we were able to focus our analytical efforts on the most impactful variables. This targeted approach not only streamlined our modeling process but also provided clear insights into the pathophysiological links between OSA and CVD, offering potential avenues for targeted interventions. Subsequently, our approach distinguishes itself by employing a multi-level phenotypic LGMM analysis that not only identifies but also quantitatively evaluates the impact of diverse clinical, demographic, and physiological factors over time. Moreover, the utilization of the t-SNE and GMM approaches facilitates the categorization of patient data into distinct clusters, revealing significant patterns that delineate varying risk levels and progression pathways of CVDs among OSA patients. This clustering not only reflects the heterogeneity of the patient population but also enhances the specificity and applicability of the predictive models, ensuring that they are tailored to real-world clinical scenarios and can effectively inform targeted intervention strategies.

In our comprehensive analysis, we employed a methodical approach involving tree-based feature selection followed by the LGMM. The feature selection process was pivotal, isolating 20 key predictors that bridge direct and indirect relationships between OSA and CVD. The ranked features, as presented in **Fig 2**, showed that the total cholesterol, LDL cholesterol, and diabetes medications scored the highest. The fourth important feature is *"arthritis_ynd"*, indicative of a self-reported diagnosis of arthritis. However, the observed high variability in the importance score of this feature across different CV folds underscores the complexity and heterogeneity of the underlying inflammatory processes critically linked to both CVD and OSA. Studies have shown that systemic inflammation associated with chronic conditions like arthritis can exacerbate the risk and severity of OSA, further complicating the CVD outcomes in these patients [57–61]. Notably, five of these features are related to sleep parameters, including the average level of oxygen desaturation during apnea/hypopnea events, average oxygen saturation during total sleep, OSA diagnosis, AHI during NREM sleep, and apnea treatment. These parameters are directly linked to OSA and underscore its significance in the progression of CVD, supported by extensive clinical evidence [17,62–64]. Furthermore, our feature selection illuminated the role of several indirect biomarkers such as depression-related scales [64], diabetes medication [65], and anthropometric variables [66], underscoring

their contribution to the complex interplay between OSA and CVD. These factors, often serving as confounders, enhance our understanding of the multifaceted relationship between chronic conditions and cardiovascular risks. The comparative efficacy of the LGMM over LR in our study highlights the advantages of integrating random effects to account for temporal variations across patient visits. This is crucial as it captures the progression and fluctuation of disease markers over time, providing a more nuanced understanding of disease trajectories than traditional models, such as Cox proportional hazards models [33–36,67] or LR models [30,68].

The longitudinal LGMM analysis of our data reveals a notable improvement in the predictive capabilities of our models across successive visits, culminating in the most robust predictions at the third visit. This trend underscores the integral role of temporal dynamics in understanding CVDs progression among OSA patients. The increasing accuracy of our predictions over time likely stems from the progressive alignment of patient risk profiles and the cumulative impact of risk factors throughout the 6–10-year follow-up period [63]. Such findings echo the patterns observed in prior longitudinal research, which consistently links OSA with elevated CVD risk. For instance, the critical role of visit-to-visit blood pressure variability was determined to be a predictor for all-cause mortality, CVD incidence, and mortality [69], thus reinforcing the significance of longitudinal monitoring in this patient population. Our analysis further delineates two distinct phenotypes within the patient cohort, characterized by their unique CVD risk profiles. Cluster 2 poses a greater challenge for predictive modeling, attributed to its members' extensive comorbidity burden, more severe CVD manifestations, and a broader variability in risk factors. This complexity is indicative of the heterogeneous nature of CVD, which is influenced by a range of factors including metabolic conditions, lifestyle, and genetic predisposition. The clear demarcation between patients with and without CVD—based on pivotal variables like sleep apnea severity, lipid levels, diabetes status, and age—mirrors previous findings on hypertension risk [70]. A constellation of factors was identified, including prehypertension, macroalbuminuria, and obesity, which significantly amplify the risk of hypertension, a known precursor to CVD. This nuanced understanding of risk factor interplay is critical for tailoring intervention strategies that effectively address the multifaceted nature of CVD in OSA patients.

Our comprehensive analysis employing the t-SNE and GMM models has provided a profound insight into the complex trajectories and phenotypic clustering of CVD progression. The transitions observed in the t-SNE mappings across the three visits, spanning 6–10-years, were particularly telling of the dynamic nature of CVD in the context of OSA, revealing not just the evolution of individual patient conditions but also the emergence of distinct, identifiable phenotypes that evolve through the clinical timeline. Subsequent application of GMM to these t-SNE outputs allowed for rigorous statistical modeling of these phenotypic clusters. This method effectively quantified the probability distributions of belonging to a particular cluster, thus enabling a more nuanced understanding of patient groupings based on shared characteristics and disease trajectories. Critical biomarkers such as triglycerides and hip circumference highlighted by the KL divergence, which are well-documented in their association with CVD risks [71,72], showed significant variability across clusters. For example, our analysis linked higher triglyceride levels—associated with severe atherosclerotic conditions—to specific clusters, underscoring their role in advancing CVD pathology [73,74]. This correlation is crucial as it aligns with emerging research suggesting that not just the presence of triglycerides, but their interaction with other metabolic factors, defines their impact on cardiovascular health. Similarly, disparities in hip circumference, which have been shown to predict cardiovascular outcomes differently based on gender, were instrumental in distinguishing between clusters, suggesting that body composition metrics can be predictive markers of CVD risk and outcomes.

Moreover, the variable importance of sleep-related parameters such as the severity of hypopneas and the degree of nocturnal oxyhemoglobin desaturation was critical in delineating clusters. These findings corroborate the significant body of literature indicating the detrimental impact of disrupted sleep architecture on cardiovascular health, thereby reinforcing the integrated approaches in managing patients with OSA and CVDs [75–77]. Mechanistically, the interplay between OSA severity, vascular dysfunction, and metabolic dysregulation provides insight into the observed CVD progression patterns. Recurrent intermittent hypoxia induces oxidative stress and systemic inflammation, leading to endothelial dysfunction and

arterial stiffness [78], while sleep fragmentation disrupts glucose homeostasis through increased insulin resistance [79]. These synergistic pathways – vascular inflammation, sympathetic overactivity, and metabolic dysfunction – may explain the aggressive disease trajectory in Cluster 2. In particular, patients in Cluster 2 likely experienced heightened sympathetic activation and severe nocturnal hypoxemia, intensifying the inflammatory cascade and metabolic disruption, which resulted in their observed rapid onset and deterioration of cardiovascular conditions compared to Cluster 1.

Our proposed research, which combines the LGMM for patient trajectory analysis and clustering using t-SNE and GMM, holds significant potential for integration into clinical workflows. Specifically, the Logistic Mixed-Effects Model could be embedded into electronic health records (EHR) utilizing 20 selected features to provide predictions of CVD outcomes, enabling clinicians to dynamically track patient trajectories and make timely interventions. t-SNE and GMM-based clustering results could further classify patients into phenotypic groups directly within EHR systems, offering a visual and data-driven understanding of patient profiles. Our clustering analysis identifies two distinct patient groups with direct implications for clinical decision-making:

- *Cardiovascular stability cluster* (Cluster 1): This cluster includes patients with more favorable health profiles, characterized by lower LDL cholesterol, fasting glucose, and fewer episodes of nocturnal hypoxia, indicating stable disease progression and effective management of CVD and OSA. These findings suggest the importance of maintaining current treatment regimens and focusing on preventive care, such as lifestyle modifications and routine monitoring. Patients in Cluster 1, characterized by slower CVD progression, may benefit from more conservative treatment approaches, with careful monitoring of CPAP adherence and standard cardiovascular risk management [80].

- *Cardiovascular risk cluster* (Cluster 2): This cluster includes patients with higher values of critical CVD biomarkers and more severe OSA symptoms, reflecting rapid disease progression and a need for intensive management strategies. For these patients, more aggressive lipid-lowering therapies, stricter glucose control, and potentially earlier initiation of lipid-lowering medications [81] as well as targeted interventions for sleep apnea, such as CPAP therapy, may be necessary.

These clustering insights support precision medicine by stratifying patients based on risk, enhancing decision-support tools to flag high-risk phenotypes, suggesting personalized interventions, and prioritizing resource allocation. For instance, patients identified in high-risk clusters could automatically trigger alerts or recommendations for further diagnostic tests or follow-up care. Monitoring nocturnal hypoxia, defined as oxygen saturation below 90% of sleep time, can also aid in identifying high-risk individuals, as prolonged nocturnal desaturation is associated with increased risks of CVD [82]. OSA screening is recommended for patients with poorly controlled hypertension, heart failure, recurrent atrial fibrillation, or a history of stroke, with CPAP therapy prioritized for severe OSA cases. Integrating risk stratification and treatment strategies into routine cardiovascular practice not only supports personalized care but also addresses the under-recognition of OSA and its critical role in CVD outcomes.

Furthermore, the temporal progression patterns identified in our study could inform the timing and intensity of interventions, allowing clinicians to proactively adjust treatment plans based on a patient's phenotypic classification. Building on these insights, the analysis of S5 and S6 Tables provides the following actionable recommendations for patients in Group 7 (long-term CVD):

- **Cluster 1 recommendations**: Patients in Cluster 1 should focus on maintaining their current metrics within a healthy range, including lipid levels, AHI, and NREM-AHI. Regular check-ups and proactive management of these indicators are important to prevent a potential transition to Cluster 2.

- **Cluster 2 recommendations**: Patients in Cluster 2 should undergo regular monitoring and targeted medical interventions to control lipid levels and improve respiratory health. To facilitate the transition to Cluster 1, patients should focus on comprehensive lifestyle changes, including adopting a heart-healthy diet, increasing physical activity, and managing

stress, as well as adopting specific treatments as needed for cholesterol management, weight loss, or respiratory metrics. We recommend that patients aim for the following target values: total cholesterol between 123.50 mg/dL and 155.33 mg/dL, with a target of 139.42 mg/dL; LDL between 56.25 mg/dL and 81 mg/dL, with a target of 68.63 mg/dL; triglycerides between 102.00 mg/dL and 108.33 mg/dL, with a target of 105.17 mg/dL; and AHI between 12.93 and 22.60, with a target of 17.77.

Recent research indicates that extremely low or high LDL-C levels might be linked to higher mortality rates. LDL levels below 50 mg/dL or above 130 mg/dL have been linked to a higher risk of mortality in patients with coronary artery disease [83], highlighting the relationship between extreme LDL levels and all-cause death rates. Additionally, severe OSA (AHI ≥ 30) has been associated with a higher risk of both all-cause and cardiovascular mortality [3]. However, it is important to note that the baseline characteristics are not fully aligned across the sub-groups. This heterogeneity limits our ability to define specific thresholds for clusters or to provide definitive recommendations regarding remaining in or transitioning to a preferable healthy cluster. Consequently, while we can offer general suggestions on managing key indicators, such as lipid levels and respiratory health based on our findings, further studies with more consistent baseline data are needed to establish clearer thresholds and provide precise recommendations for achieving optimal cluster outcomes.

Our study acknowledges several limitations that warrant discussion and guide future research directions. Firstly, the complexity and heterogeneity of the OSA-CVD interplay may have introduced variability in our model predictions. While we utilized advanced statistical techniques to model this complex relationship longitudinally, the intrinsic variability of individual patient responses, especially regarding intervention outcomes, might not have been fully captured. This could affect the generalizability of our findings to broader populations. Although SMOTE helps in addressing the class imbalance inherent in many clinical datasets, the technique generates synthetic data points that may not perfectly represent real-world conditions. Consequently, there is a risk that models trained on this data could be biased towards the over-represented class. To mitigate this risk, our analysis was complemented by stringent cross-validation during model evaluation. The data used for this phase did not include SMOTE-processed samples, ensuring that our evaluation metrics are based on actual clinical outcomes rather than synthetic approximations. Nonetheless, the initial use of SMOTE for feature selection could still influence the final model by potentially enhancing certain patterns that are not as pronounced in real-world data.

Moreover, our study faced challenges with data attrition and irregular follow-up intervals, common in longitudinal studies, which could introduce bias or affect the robustness of the results. Additionally, while our models incorporated numerous relevant variables, they might not account for all possible confounders, such as socioeconomic status or environmental factors, which could influence the progression and management of CVD and OSA. The absence of key lifestyle factors, such as smoking, alcohol use, and caffeine intake, due to the feature selection process, may have resulted in unaccounted variability in the data, potentially obscuring the true relationships between the observed features and cardiovascular risk. Given the nature of the current dataset, it was not possible to address these confounders comprehensively. Their potential impact on the clustering process, including altering the composition and interpretation of the identified phenotypic clusters, and limiting the model's ability to fully capture the multifactorial etiology of cardiovascular conditions, should not be overlooked. Future research should focus on integrating more comprehensive datasets that include these lifestyle factors and employing causal inference methods to address confounding more robustly. Techniques such as structural equation modeling (SEM) or propensity score adjustment could help isolate the effects of these unmeasured variables, providing a more nuanced understanding of the phenotypic clusters and their relationship with cardiovascular outcomes.

While our findings demonstrate strong predictive capability in the WSCS, we also acknowledge important limitations in generalizability. Our study population, predominantly middle-aged and demographically homogeneous, may not fully represent the spectrum of CVD-OSA manifestations across diverse populations. Although our models achieved 80.5% diagnostic accuracy in predicting CVD outcomes, several key aspects require further validation. First, the model's performance needs to be evaluated in younger patients (<40 years) who may present with different early markers of CVD

progression, and in elderly populations (>75 years) who often have multiple competing comorbidities. Second, our clinical indicators analysis, which identified total cholesterol, LDL, and diabetes as key predictors, requires validation in populations with different genetic predispositions, varying baseline metabolic profiles, and diverse dietary and lifestyle patterns that may influence both CVD and OSA progression. The transition patterns observed between Cluster 1 (slower CVD progression) and Cluster 2 (rapid onset/deterioration) also need to be verified across populations with different socioeconomic barriers to healthcare access and in healthcare systems with varying approaches to CVD and OSA management. Future multi-center studies should evaluate these phenotypic clusters in populations with different baseline autonomic nervous system characteristics and environmental exposures that may affect sleep patterns, ensuring the broader clinical applicability of our phenotypic modeling approach. Moreover, integrating novel biomarkers and more detailed genetic information could improve the predictive power of the models and offer deeper insights into the biological underpinnings of the disease processes.

Our current work relies on self-reported data, particularly for variables related to medical history, which are subjected to recall bias and may impact the accuracy of the findings. Given that the association between OSA and CVD extends to several comorbidities, including obesity, dyslipidemia, and diabetes, incorporating data on metabolic syndrome biomarkers, such as Lipid Accumulation Product (LAP), Visceral Adiposity Index (VAI), Atherogenic Index of Plasma (AIP), and the insulin resistance marker TyG index, could enhance our understanding of OSA-CVD progression and its predictors [84,85]. Considering our findings, it is imperative to acknowledge the limitations associated with data scarcity for Group 1 and Group 3. While our analysis provides initial insights into the disease progression and interactions within these groups (as illustrated in **Figs 4** and **5**), the conclusions drawn are provisional. Therefore, further validation using additional datasets is essential to confirm these findings and enhance the robustness of our predictive models. Finally, the use of real-time data collection technologies, such as wearable devices, could provide continuous, high-resolution data that better capture the dynamics of CVDs and OSA. This approach would allow for a more nuanced analysis of the interplay between sleep patterns, physiological changes, and cardiovascular health, potentially leading to more personalized and timely interventions.

## 5. Conclusion

Our study elucidates the complex interplay between OSA and CVDs through a longitudinal analysis utilizing the WSCS. Our findings not only clarify the mechanisms underlying these comorbid conditions but also enhance predictive accuracies which could significantly impact clinical practices. Firstly, our research demonstrates the potent influence of specific biomarkers such as total cholesterol, LDL, and indicators of diabetes on the progression of CVDs in the presence of OSA. These factors serve as critical predictors within our logistic mixed-effects models, offering clinicians robust tools to identify at-risk patients early in the disease progression timeline. Moreover, the phenotypic clusters identified through our analysis provide a nuanced understanding of patient risk profiles. These clusters categorize patients into distinct groups with differing risks and progression patterns of disease, enabling personalized treatment approaches. Patients in high-risk clusters may benefit from aggressive management strategies including advanced CPAP therapy and targeted lipid-lowering treatments, whereas those in more stable clusters might be managed with routine monitoring and conservative care strategies.

The practical applications of our findings are substantial and have significant implications for clinical practice. Specifically, our research emphasizes the need for tailored therapeutic strategies that specifically consider the complex dynamics between OSA and CVDs. By identifying and delineating the various phenotypic profiles associated with these comorbidities, clinicians can develop more personalized treatment plans that directly address the unique risk factors presented by individual patients. Our findings suggest that patients with severe OSA and concurrent high cardiovascular risk may benefit from an integrated treatment approach that includes aggressive management of sleep apnea with CPAP therapy, alongside proactive cardiovascular risk mitigation strategies such as statin therapy and lifestyle modifications. Conversely, patients identified with less severe forms of OSA without significant cardiovascular risk markers might be managed with

less intensive interventions, focusing on lifestyle changes and regular monitoring. Moreover, our study's approach aids in the efficient allocation of healthcare resources by enabling healthcare providers to prioritize interventions for patients who are most at risk. This targeted intervention strategy not only improves the cost-effectiveness of healthcare delivery but also maximizes patient outcomes by preventing the progression of disease with precisely timed therapeutic interventions.

## 5.1. Table of nomenclature

| Abbreviation | Definition |
|---|---|
| AHI | Apnea-Hypopnea Index |
| BIC | Bayesian Information Criterion |
| BMI | Body Mass Index |
| CI | Confidence Interval |
| CVD | Cardiovascular Disease |
| GMM | Gaussian Mixture Model |
| HDL | High-density Lipoprotein |
| KLD | Kullback-Leibler Divergence |
| KNN | K-Nearest Neighbor |
| LDL | Low-density Lipoprotein |
| LGMM | Logistic Gaussian Mixture Model |
| LR | Logistic Regression |
| MACE | Major Adverse Cardiovascular Event |
| NREM-AHI | Non-rapid Eye Movement Apnea-Hypopnea Index |
| OSA | Obstructive Sleep Apnea |
| SD | Standard Deviation |
| SE | Standard Error |
| SMOTE | Synthetic Minority Over-sampling Technique |
| t-SNE | t-distributed Stochastic Neighbor Embedding |
| WSCS | Wisconsin Sleep Cohort Study |

## Supporting information

**S1 Table. Description of variables.**
(DOCX)

**S2 Table. Classification based on total cholesterol, triglyceride, AHI and BMI.**
(DOCX)

**S3 Table. Comparative analysis of variables for healthy patients from Visit 2 who tend to move to Cluster 1 or Cluster 2 in Visit 3 within Group 1.**
(DOCX)

**S4 Table. Comparative analysis of variables for healthy patients from Visit 1 who tend to move to Cluster 1 or Cluster 2 in Visit 2 within Group 3.**
(DOCX)

**S5 Table. Comparative analysis of variables for patients from Visit 1 who tend to move to Cluster 1 or Cluster 2 in Visit 2 within Group 7.2 and Group 7.3.**
(DOCX)

**S6 Table. Comparative analysis of variables for patients from Visit 2 who tend to move to Cluster 1 or Cluster 2 in Visit 3 within Group 7.2 and Group 7.3.**
(DOCX)

## Author contributions

**Conceptualization:** Duy Nguyen, Ca Hoang, Abhay Sharma, Trung Quoc Le, Phat Kim Huynh.

**Data curation:** Duy Nguyen, Ca Hoang, Tien Truong.

**Formal analysis:** Duy Nguyen, Ca Hoang, Tien Truong.

**Investigation:** Duy Nguyen, Ca Hoang, Dang Nguyen, Hillary Gia Lam, Abhay Sharma, Trung Quoc Le, Phat Kim Huynh.

**Methodology:** Duy Nguyen, Ca Hoang, Tien Truong, Trung Quoc Le, Phat Kim Huynh.

**Project administration:** Phat Kim Huynh.

**Resources:** Trung Quoc Le.

**Supervision:** Trung Quoc Le, Phat Kim Huynh.

**Validation:** Duy Nguyen, Ca Hoang, Abhay Sharma, Trung Quoc Le, Phat Kim Huynh.

**Visualization:** Duy Nguyen, Ca Hoang, Tien Truong, Trung Quoc Le, Phat Kim Huynh.

**Writing – original draft:** Duy Nguyen, Ca Hoang, Tien Truong, Dang Nguyen, Hillary Gia Lam, Trung Quoc Le, Phat Kim Huynh.

**Writing – review & editing:** Duy Nguyen, Ca Hoang, Trung Quoc Le, Phat Kim Huynh.

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
