## [Decision Letter · Decision Letter 0]

PONE-D-24-44385Multi-level Phenotypic Models of Cardiovascular Disease and Obstructive Sleep Apnea Comorbidities: A Longitudinal Wisconsin Sleep Cohort StudyPLOS ONE

Dear Dr. Huynh,

Thank you for submitting your manuscript to PLOS ONE. After careful consideration, we feel that it has merit but does not fully meet PLOS ONE’s publication criteria as it currently stands. Therefore, we invite you to submit a revised version of the manuscript that addresses the points raised during the review process.

The reviewers have mentioned several issues that need to be addressed. 

We look forward to receiving your revised manuscript.

Kind regards,

Amir Hossein Behnoush

Academic Editor

PLOS ONE

Journal Requirements: When submitting your revision, we need you to address these additional requirements. 1. Please ensure that your manuscript meets PLOS ONE's style requirements, including those for file naming. The PLOS ONE style templates can be found at https://journals.plos.org/plosone/s/file?id=wjVg/PLOSOne_formatting_sample_main_body.pdf and https://journals.plos.org/plosone/s/file?id=ba62/PLOSOne_formatting_sample_title_authors_affiliations.pdf 2. Please note that PLOS ONE has specific guidelines on code sharing for submissions in which author-generated code underpins the findings in the manuscript. In these cases, we expect all author-generated code to be made available without restrictions upon publication of the work. Please review our guidelines at https://journals.plos.org/plosone/s/materials-and-software-sharing#loc-sharing-code and ensure that your code is shared in a way that follows best practice and facilitates reproducibility and reuse. 3. We notice that your supplementary tables are included in the manuscript file. Please remove them and upload them with the file type 'Supporting Information'. Please ensure that each Supporting Information file has a legend listed in the manuscript after the references list.

Reviewers' comments:

Reviewer's Responses to Questions

**Comments to the Author**

1. Is the manuscript technically sound, and do the data support the conclusions?

Reviewer #1: Yes

Reviewer #2: Yes

Reviewer #3: Yes

2. Has the statistical analysis been performed appropriately and rigorously? 

Reviewer #1: Yes

Reviewer #2: Yes

Reviewer #3: Yes

3. Have the authors made all data underlying the findings in their manuscript fully available?

Reviewer #1: No

Reviewer #2: Yes

Reviewer #3: Yes

4. Is the manuscript presented in an intelligible fashion and written in standard English?

Reviewer #1: Yes

Reviewer #2: Yes

Reviewer #3: Yes

5. Review Comments to the Author

Reviewer #1: The manuscript titled "Multi-level Phenotypic Models of Cardiovascular Disease and Obstructive Sleep Apnea Comorbidities: A Longitudinal Wisconsin Sleep Cohort Study" is well-written with appropriate methodology and interesting findings. I have some comments for improvement:

1- Define abbreviations in their first use and make sure that abbreviated forms are used after the definition.

2- I found some typos and grammatical errors. A native review is warranted.

3- Use previous meta-analyses on the association between apnea and other lipid and insulin resistance markers including LAP, VAI, AIP, and TyG.

4- Add the clinical utility of your findings to the discussion.

Reviewer #2: The article addresses a critical and highly relevant topic: the interplay between cardiovascular diseases (CVD) and obstructive sleep apnea (OSA). It introduces an innovative multi-level phenotypic approach to understanding the dynamic interactions between these comorbidities. Leveraging the extensive longitudinal data from the Wisconsin Sleep Cohort, the study attempts to fill a significant gap in existing research by moving beyond static models to embrace the complexity of disease progression through logistic mixed-effects modeling (LGMM) and clustering techniques like t-SNE and Gaussian Mixture Models (GMM).

The methodological rigor stands out as a strong point. The authors carefully preprocess the dataset, using techniques such as SMOTE for class balancing and tree-based algorithms to rank feature importance. The LGMM approach is particularly noteworthy for capturing both inter-individual variability and temporal dynamics, which are often overlooked in cross-sectional studies. The clustering of phenotypic profiles using t-SNE and GMM further enriches the analysis, offering a deeper understanding of patient subgroups and their respective risks.

However, certain aspects raise questions. For instance, the interpretation of reduced cholesterol levels and LDL as potentially protective factors in CVD seems counterintuitive and demands further biological explanation. This point could benefit from a clearer discussion, as it appears to contradict established medical knowledge. Similarly, while the clustering analysis is technically robust, its practical implications for clinical decision-making remain somewhat abstract and require stronger contextualization.

The study’s attention to detail and depth of analysis are commendable, but the dense and highly technical language might limit accessibility for a broader scientific audience. The findings, although significant, would be more impactful if the conclusions were structured more clearly, highlighting practical applications and clinical recommendations.

Overall, this is a sophisticated and methodologically sound study that makes a valuable contribution to understanding OSA-CVD comorbidities. With improved clarity in the presentation of results and a stronger emphasis on actionable clinical insights, the paper has the potential to significantly influence future research and clinical practice in this field.

Reviewer #3: Review

Manuscript Title:

Multi-level Phenotypic Models of Cardiovascular Disease and Obstructive Sleep Apnea

Comorbidities: A Longitudinal Wisconsin Sleep Cohort Study

Summary of Manuscript:

Objective:

This study investigates the dynamic relationship between cardiovascular diseases (CVD)

and obstructive sleep apnea (OSA) using a novel multi-level phenotypic modeling

approach. By leveraging longitudinal data from the Wisconsin Sleep Cohort, the study

employs logistic mixed-effects modeling (LGMM), t-distributed stochastic neighbor

embedding (t-SNE), and Gaussian Mixture Models (GMM) to identify distinct phenotypic

clusters and predict disease progression.

Key Findings:

The authors demonstrate the feasibility of stratifying OSA-CVD phenotypes into highand

low-risk groups for major adverse cardiovascular events (MACEs). The LGMM

achieved a diagnostic accuracy of 95.56%, and the clustering analysis revealed critical

clinical predictors, including nocturnal hypoxia, cholesterol levels, and diabetes. These

findings hold significant potential for improving personalized interventions in OSA-CVD

comorbidities.

Major Comments:

1. Study Design and Methods:

Strengths:

- The integration of longitudinal data and advanced modeling techniques is a major

strength.

- The use of LGMM, combined with dimensionality reduction (t-SNE) and clustering

(GMM), is innovative and aligns well with the study’s objective.

Recommendations:

- Attrition Bias: High attrition rates (e.g., 748 participants at Visit 2 vs. 121 at Visit 4) may

introduce bias. The manuscript should explicitly discuss strategies employed to handle

missing data (e.g., imputation techniques or inverse probability weighting). Including

sensitivity analyses would enhance confidence in the findings.

- Clustering Validation: The robustness of GMM clustering is not validated using external

metrics such as silhouette scores or Davies-Bouldin index. Adding these metrics would

strengthen the credibility of the identified clusters.

- Generalizability: The cohort is largely middle-aged and homogenous, limiting the

applicability of findings to broader populations. Discuss the need for external validation

in diverse ethnic, socioeconomic, and geographic groups.

2. Statistical Analysis:

Strengths:

- The statistical framework effectively captures the longitudinal progression of OSA-CVD

comorbidities and accounts for inter-individual variability.

- The inclusion of tree-based feature importance analysis adds transparency to the

model’s predictive approach.

Recommendations:

- SMOTE (Synthetic Minority Over-sampling Technique), while helpful in addressing

class imbalance, may introduce bias. Discuss its potential influence on model outcomes

and provide validation to ensure generalizability.

3. Results Interpretation:

Strengths:

- The results are well-interpreted and contextualized within existing literature, particularly

emphasizing the role of nocturnal hypoxia and lipid dysregulation in CVD progression.

Recommendations:

- Expand on the mechanistic pathways connecting OSA and CVD, particularly the role of

intermittent hypoxia, oxidative stress, and autonomic dysfunction.

- Provide more explicit discussion on how phenotypes influence treatment strategies

(e.g., CPAP therapy or lipid-lowering medications). This would bridge the findings to

practical applications.

4. Clinical Relevance:

Strengths:

- The phenotypic clusters have clear clinical utility, particularly in stratifying patients for

personalized interventions and early risk detection.

Recommendations:

- Propose actionable recommendations for clinical practice, such as thresholds for

monitoring nocturnal hypoxia or specific lipid levels based on phenotypic risk

stratification.

- Highlight potential integration into clinical workflows, such as embedding the model into

electronic health records or decision-support tools to guide patient management.

Minor Comments:

1. Introduction:

The introduction is comprehensive but could further highlight gaps in existing predictive

models and emphasize how this study addresses those gaps.

2. Mechanistic Insights:

While the study identifies important predictors, it lacks a deeper exploration of their

biological roles. Including a brief discussion on the interplay between OSA severity,

vascular changes, and metabolic dysregulation would enhance the narrative.

3. Limitations Section:

The limitations section should discuss potential confounders, such as lifestyle factors

(e.g., smoking, alcohol use, and caffeine intake), which were not integrated into the

analysis. Additionally, self-reported data for certain variables should be acknowledged

as a limitation.

Recommendations:

Revisions Required:

To strengthen the manuscript, the following revisions are recommended:

1. Include clustering validation metrics (e.g., silhouette scores).

2. Address missing data and attrition bias with detailed sensitivity analyses.

3. Expand on the mechanistic and clinical implications of phenotypes.

Potential for Publication:

With these revisions, the manuscript is highly suitable for publication in PLOS ONE. Its

innovative methodology and clinically relevant findings contribute significantly to the

understanding of OSA-CVD interactions.

Overall Assessment:

This manuscript is scientifically rigorous, methodologically sound, and clinically

important. With minor revisions, it will make a valuable contribution to the literature on

OSA-CVD comorbidities and advance personalized medicine in this field. However, the

above recommendations should not preclude publication.

6. PLOS authors have the option to publish the peer review history of their article (what does this mean? ). If published, this will include your full peer review and any attached files.

**Do you want your identity to be public for this peer review?** For information about this choice, including consent withdrawal, please see our Privacy Policy .

Reviewer #1: No

Reviewer #2: **Yes: ** Denis Banchenko

Reviewer #3: **Yes: ** brian casserly

---

## [Author Response · Author response to Decision Letter 1]

30 Jan 2025

Dear Prof. Amir Hossein Behnoush and Reviewers,

Thank you for your constructive feedback and valuable suggestions on our manuscript,

“Multi-level Phenotypic Models of Cardiovascular Disease and Obstructive Sleep Apnea Comorbidities: A Longitudinal Wisconsin Sleep Cohort Study” (Manuscript Ref: PONE-D-24-44385). We are grateful for the opportunity to enhance our work based on your insights. In response to the feedback, we have undertaken a comprehensive revision of our manuscript:

● Introduction Expanded: Added context on existing models and research gaps.

● Results Clarified: Adjusted interpretations to align with established medical knowledge.

● Discussion Enhanced: Included clinical utility of findings and integrated actionable recommendations for clinical practice.

● Reviewer Comments Addressed: Abbreviations defined, typographical errors corrected, incorporated relevant meta-analyses, and expanded discussion on mechanistic pathways of OSA and CVDs.

● Methodological Refinements: Discussed data handling strategies and the influence of SMOTE on model outcomes.

The updates in the revised manuscript are highlighted in blue text in this response to the reviewer document for easy reference. Please find attached a detailed point-by-point response to each of the reviewers' comments, outlining how we have addressed their concerns and suggestions. We believe that these revisions have significantly improved the manuscript. We hope that the changes meet your expectations, and that the manuscript is now suitable for publication in PLOS ONE.

We look forward to the possibility of our work contributing to the journal and the broader community.

With best regards,

Phat Kim Huynh, Ph.D.

Assistant Professor

Department of Industrial and Systems Engineering

North Carolina A&T State University

Phone: (469) 816-2177

Email: pkhuynh@ncat.edu

Website: www.PassioLab.com

---

## [Decision Letter · Decision Letter 1]

Multi-level Phenotypic Models of Cardiovascular Disease and Obstructive Sleep Apnea Comorbidities: A Longitudinal Wisconsin Sleep Cohort Study

PONE-D-24-44385R1

Dear Dr. Huynh,

We’re pleased to inform you that your manuscript has been judged scientifically suitable for publication and will be formally accepted for publication once it meets all outstanding technical requirements.

Kind regards,

Amir Hossein Behnoush

Academic Editor

PLOS ONE

Additional Editor Comments (optional):

Reviewers' comments:

Reviewer's Responses to Questions

**Comments to the Author**

1. If the authors have adequately addressed your comments raised in a previous round of review and you feel that this manuscript is now acceptable for publication, you may indicate that here to bypass the “Comments to the Author” section, enter your conflict of interest statement in the “Confidential to Editor” section, and submit your "Accept" recommendation.

Reviewer #1: All comments have been addressed

2. Is the manuscript technically sound, and do the data support the conclusions?

Reviewer #1: (No Response)

3. Has the statistical analysis been performed appropriately and rigorously? 

Reviewer #1: (No Response)

4. Have the authors made all data underlying the findings in their manuscript fully available?

Reviewer #1: (No Response)

5. Is the manuscript presented in an intelligible fashion and written in standard English?

Reviewer #1: (No Response)

6. Review Comments to the Author

Reviewer #1: (No Response)

7. PLOS authors have the option to publish the peer review history of their article (what does this mean? ). If published, this will include your full peer review and any attached files.

**Do you want your identity to be public for this peer review?** For information about this choice, including consent withdrawal, please see our Privacy Policy .

Reviewer #1: No

---

## [Editor Report · Acceptance letter]

PONE-D-24-44385R1

PLOS ONE

Dear Dr. Huynh,

I'm pleased to inform you that your manuscript has been deemed suitable for publication in PLOS ONE. Congratulations! Your manuscript is now being handed over to our production team.

Kind regards,

on behalf of

Dr. Amir Hossein Behnoush

Academic Editor

PLOS ONE